# Plasticity of *Drosophila* germ granules during germ cell development

**Anna C. Hakes, Elizabeth R. Gavis** *

Department of Molecular Biology, Princeton University, Princeton, New Jersey, United States of America

* gavis@princeton.edu

## Abstract

Compartmentalization of RNAs and proteins into membraneless structures called granules is a ubiquitous mechanism for organizing and regulating cohorts of RNAs. Germ granules are ribonucleoprotein (RNP) assemblies required for germline development across the animal kingdom, but their regulatory roles in germ cells are not fully understood. We show that after germ cell specification, *Drosophila* germ granules enlarge through fusion and this growth is accompanied by a shift in function. Whereas germ granules initially protect their constituent mRNAs from degradation, they subsequently target a subset of these mRNAs for degradation while maintaining protection of others. This functional shift occurs through the recruitment of decapping and degradation factors to the germ granules, which is promoted by decapping activators and renders these structures P body-like. Disrupting either the mRNA protection or degradation function results in germ cell migration defects. Our findings reveal plasticity in germ granule function that allows them to be repurposed at different stages of development to ensure population of the gonad by germ cells. Additionally, these results reveal an unexpected level of functional complexity whereby constituent RNAs within the same granule type can be differentially regulated.

## Introduction

Ribonucleoprotein (RNP) granules are biomolecular condensates containing RNAs and RNA-binding proteins that create cytoplasmic compartments without the use of membranes. Compartmentalization in granules concentrates RNAs together with regulatory proteins, such as those involved in RNA localization, translational control, RNA processing, and control of mRNA stability [1–3]. Thus, RNP granules are hypothesized to be hubs of posttranscriptional regulation. Several types of RNP granules—such as processing bodies (P bodies), stress granules, and neuronal transport granules—are found in many different cell types and species, suggesting a common and highly conserved regulatory strategy [1,4].

One class of RNP granules, called germ granules, is a characteristic feature of germ cells across animal species [5]. Germ granules contain mRNAs and proteins that are necessary for germ cell development and are therefore thought to play important roles in germline development and function [6,7]. In some animals, like *Drosophila*, *Xenopus*, and zebrafish, germ granules form during oogenesis from maternally expressed proteins and RNAs. During early

**Data Availability Statement:** All relevant data are within the paper and its Supporting Information files.

**Funding:** This work was funded by National Institute of Health (NIH) grant R35 GM126967 to ERG. ACH was supported by NIH training grant

T32 GM007388. The funders had no role in study design, data collection and analysis, decision to publish, or preparation of the manuscript.

**Competing interests:** The authors have declared that no competing interests exist.

**Abbreviations:** CHX, cycloheximide; MZT, maternal to zygotic transition; RNP, ribonucleoprotein; smFISH, single-molecule fluorescence in situ hybridization; STED, stimulated emission depletion.

embryogenesis, these maternally supplied granules are segregated to a subset of cells that will give rise to the germline. In other animals, including mammals, germ granules are not acquired maternally, but similar granules containing necessary germline determinants form de novo after the germ cells have been specified [5,8]. Despite differences in how germ granules arise, shared components like the RNA-binding protein Nanos (Nos), Tudor (Tud) domain proteins, and RNA helicases [5,9,10] suggest a conserved role in RNA metabolism.

In *Drosophila*, the germ granules form at the posterior of the oocyte within a specialized cytoplasm called the germ plasm. There, the germ granule proteins Oskar (Osk), Vasa (Vas), Tud, and Aubergine (Aub) form a scaffold [11–13] to recruit mRNAs that are necessary for germ cell development, including *nanos* (*nos*), *polar granule component* (*pgc*), and *Cyclin B* (*CycB*) [14–16]. Within the germ granules, these RNAs self-associate to form spatially distinct homotypic clusters [17–19]. The germ granules become anchored to the posterior cortex by the end of oogenesis and persist there into embryogenesis.

During embryogenesis, the germ plasm induces the formation of membrane buds around nuclei at the posterior of the syncytial embryo, which then pinch off to form the germ cell progenitors—called pole cells [20]. The germ granules accumulate around these nuclei and their associated centrosomes by dynein-dependent transport. Co-packaging of many different mRNAs into each granule ensures efficient trafficking and segregation of mRNAs to the pole cells as they bud and divide [21]. Germ granules also play a role in stabilizing constituent RNAs during the maternal to zygotic transition (MZT), when a majority of maternal mRNAs are degraded in the somatic region of the embryo. This degradation allows zygotic transcription to be activated, which is necessary for cellularization and differentiation of the soma and for body patterning [22,23]. In contrast, numerous transcripts, including important germline determinants, are stabilized in the germ plasm [23–25]. Although not all stabilized mRNAs are localized to the germ granules, sequestration within germ granules may be a mechanism to stabilize a subset of these RNAs by making them less accessible to mRNA decay factors.

After the pole cells have formed and their contents are physically separated from the soma, the need for protection from the somatic MZT is eliminated. Furthermore, once the pole cells enter mitotic quiescence in advance of gastrulation [26], the role of germ granules in distributing mRNAs to daughter cells is no longer required. However, the germ granules persist into later stages of embryogenesis [27]. Interestingly, germ granules increase in size in the pole cells [28–30]. The significance of this morphological change and what roles the germ granules play throughout the remainder of germ cell development have yet to be determined. Since germ granules are a conserved feature of differentiated germ cells, deciphering their regulatory functions at these stages is of particular interest.

Here, we have investigated a role for germ granules in regulating mRNA stability in pole cells. We found that during a period when germ granules gain the ability to fuse with each other, they sequentially recruit mRNA decay factors typically found in P bodies, and *nos* and *pgc* are destabilized. In contrast, *CycB* is maintained throughout embryogenesis, despite residing within the same granules. These findings suggest that in contrast to their protective role prior to pole cell formation, germ granules play a more complex role in pole cells, selectively protecting some mRNAs while promoting the degradation of others. Overexpression of an activating subunit of the decapping complex, Decapping protein 1 (DCP1), disrupts the protection of *CycB*, suggesting DCP1 levels are limiting for *CycB* degradation. We show that the decapping activators Edc3 and Patr-1 are necessary to localize the decapping complex to germ granules and disruption of decapping complex recruitment leads to aberrant stabilization of *nos* and *pgc*. Furthermore, disrupting the selective mRNA protection or degradation by germ granules leads to defects in pole cell migration, suggesting both of these germ granule functions are necessary for proper gonad formation. Overall, these findings reveal a shift in germ

granule function after pole cell formation that is required for proper mRNA regulation and pole cell development. Such plasticity allows the same RNP granules to be repurposed for distinct functions at different stages of development.

## Results

### Germ granules enlarge and persist in pole cells through embryogenesis

Within the pole cells, germ granules become larger and undergo shape changes [28,29,31]. To determine precisely when germ granules grow in size and how long they persist, we visualized germ granules throughout embryogenesis using Osk as a marker. To ensure that changes we observed in the germ granules reflect their normal physiology, we used CRISPR-Cas9 genome editing to endogenously tag Osk with sfGFP at its C terminus. Individuals homozygous for the endogenously tagged Osk-sfGFP are fertile and show no phenotypic abnormalities, indicating that the protein is fully functional.

We visualized Osk-sfGFP by anti-GFP immunofluorescence, using somatic nuclear cycles (nc) [32] or Bownes stages [33] to measure developmental time, starting with pole cell budding at nc9 and ending when the pole cells coalesce in the gonad (Bownes stage 14). In addition to marking the germ granules, endogenously tagged Osk is also present in nuclear puncta beginning at nc10 (Fig 1B) as previously reported for transgenically expressed Osk-GFP [30]. We

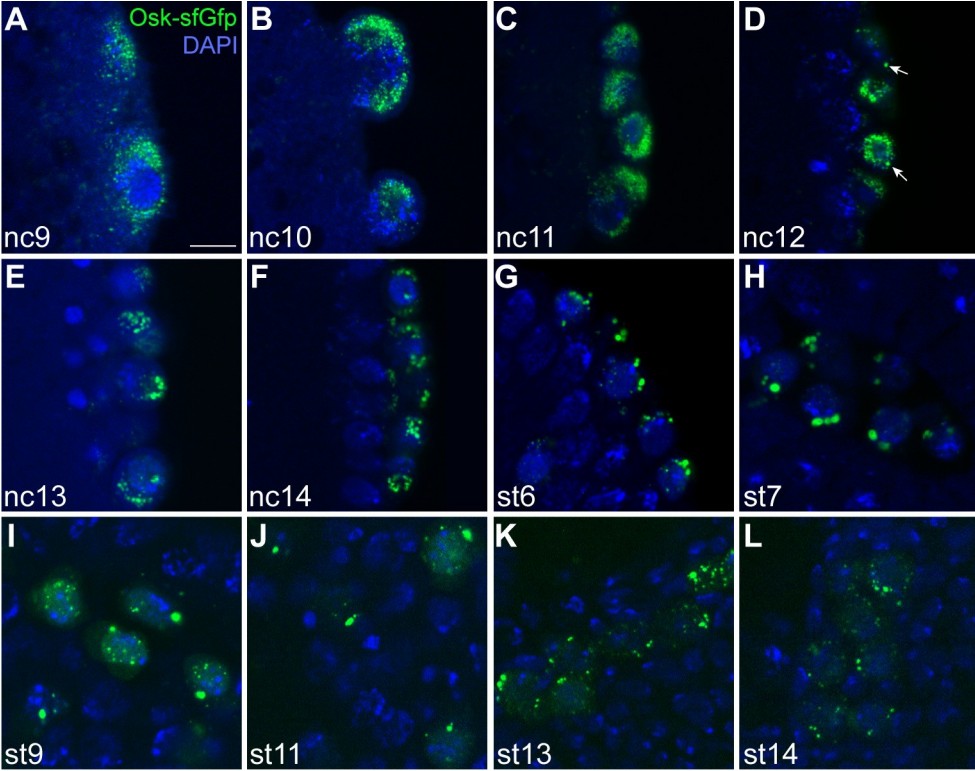

**Fig 1. Germ granules increase in size and persist through gonad formation.** Single confocal sections, showing endogenously tagged Osk-sfGFP (green) in the pole cells after pole cell budding (A–F), throughout gastrulation (G–H) and during pole cell migration (I–L). Embryos were staged by nuclear cycle (nc, A–F) or Bownes stage (st, G–L) according to nuclear density or morphological features, respectively. Osk-sfGFP was detected by anti-GFP immunofluorescence and nuclei were stained with DAPI (blue). The brightness and contrast were adjusted individually for each image to best show the features of the germ granules at that stage. Arrows indicate examples of the larger germ granules that first appear at nc12. Scale bar: 5 μm.

focused on the cytoplasmic granules, as only they contain mRNAs [30]. During nc9, these granules appear as diffraction limited spots that cluster around the budding nuclei (Fig 1A). A few larger germ granules, with diameters between 500 and 700 nm, first begin to appear in the pole cells starting at nc12, approximately 10 min after pole cell formation (Fig 1A–1D). Over the next 90 min, there is a trend toward larger granules such that by the end of nc14, most granules appear much larger than those first segregated to the pole cells. The majority of these granules are at least 1 μm in diameter, with some reaching over 1.5 μm (Fig 1E and 1F).

At the end of nc14, gastrulation begins. Cellular movements carry the pole cells into the posterior midgut primordium, where they respond to chemotactic cues directing them to migrate through the midgut epithelium and attach to adjacent mesoderm cells. Once aligned, the germline and mesodermal cells migrate and coalesce to form the gonad [34]. In the gonad, the pole cells resume cell division and ultimately generate the germline stem cells capable of producing eggs or sperm. We visualized Osk-sfGFP throughout these migratory movements to determine how long germ granules persist. Both large (≥1 μm diameter) and small granules are visible as the pole cells begin to migrate through the midgut and toward the presumptive mesoderm (Bownes stage 9). These granules persist throughout gastrulation and at least until the end of pole cell migration at stage 14 (Fig 1G–1L). Similar results were obtained using another endogenously tagged germ granule marker, Vas-EGFP [35] (S1 Fig). Their persistence suggests that the larger granules are stable and that the germ granules could play a role in germ cell development throughout embryogenesis.

### Germ granules grow through fusion in the pole cells

During this period of germ granule growth, we observed a decrease in the number of granules that coincided with their increase in size (Fig 1). Therefore, we hypothesized that germ granules enlarge via fusion of smaller granules. To test this hypothesis, we performed time lapse confocal imaging of Osk-sfGFP during nc14, when the majority of germ granule growth occurs. Throughout nc14, we observed examples of apparent fusion between 2 granules, occurring over the course of 2 to 3 min (Fig 2A–2E, S1 Video). These events are slower than fusion of the more liquid-like germ granules in *Caenorhabditis elegans*, which occurs in seconds [36].

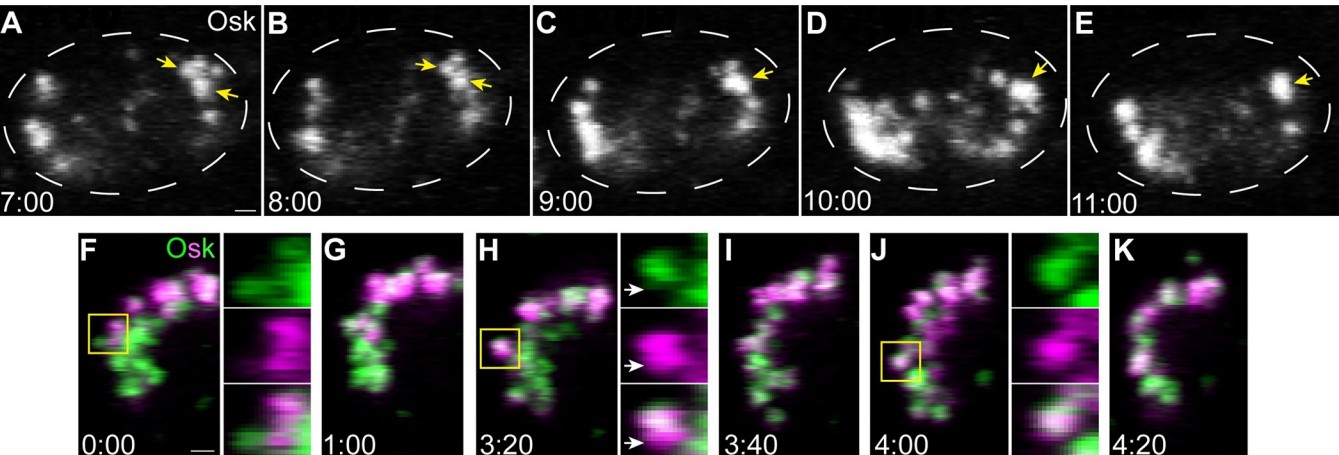

**Fig 2. Germ granules grow by fusion in the pole cells.** Maximum intensity confocal z-projections of a 1 μm region of a representative pole cell at nc14 with endogenously tagged Osk-sfGFP (A–E) or Osk-Dendra2 (F–K). Osk-sfGFP and Osk-Dendra2 images were taken from a 5-min period of S1 Video and a 4-min period of S2 Video, respectively. Yellow arrows and boxes indicate germ granules that undergo fusion. Enlargements of the boxed regions in (F), (H), and (J), show the mixing of green and red (shown here in magenta) fluorescent Osk-Dendra2 signal over time. White arrows indicate a region of a granule where the magenta labeled and green labeled contents have yet not mixed after fusion. Time stamps indicate minutes:seconds. Scale bar: 1 μm.

Furthermore, this analysis cannot distinguish true fusion from granules docking together without exchanging their materials. To confirm that fusion does occur, we endogenously tagged Osk with the photoconvertible fluorescent protein Dendra2 at its C terminus. Osk-Dendra2 was then photoconverted from green to red (shown here as green to magenta) within a small region of a pole cell to generate differentially labeled germ granules (Fig 2F) that were tracked using time lapse imaging. During nc14, we observed almost entirely photoconverted (red) and unconverted green granules that appear to "stick" together for a period of about 1 to 3 min without mixing (Fig 2F–2H, S2 and S3 Videos). After this initial period, the 2 colors begin to mix, resulting in a granule with green and red signals distributed uniformly throughout (Fig 2I–2K, S2 and S3 Videos). Together, these data suggest that germ granule growth at nc14 occurs at least in part by the slow fusion of smaller granules. Since fusion has not been observed at earlier stages [30,37], these results suggest that the germ granules are more dynamic at nc14.

## *nos* and *pgc* become destabilized in the pole cells while *CycB* is protected

To determine if enlargement of granules is accompanied by changes in their composition, we performed single-molecule fluorescence in situ hybridization (smFISH) analysis, which revealed that some of the larger granules at nc14 lack *nos* and *pgc* mRNA in contrast to earlier nuclear cycles when granules almost always contain at least 1 of these 2 mRNAs (Fig 3A). Since *nos* and *pgc* do not appear outside of germ granules at nc14, these seemingly "empty" granules suggest that the enlargement of germ granules in nc12-14 may coincide with a loss of granule mRNAs. To test this hypothesis, we measured total mRNA levels per pole cell of *nos*, *pgc*, and another abundant germ granule RNA, *CycB* [17,38], during each nuclear cycle, starting with pole cell budding at nc9. mRNAs were detected by smFISH and their levels were normalized to either endogenously tagged Osk-sfGFP or Vas-EGFP (Fig 3B) to account for the overall decrease in the amount of germ plasm per pole cell that occurs as pole cells divide prior to gastrulation.

Both *nos* and *pgc* levels decline modestly (25% and 19%, respectively) between nc9 and nc10 (Fig 3C and 3D). This decrease could be due to incomplete protection from the somatic MZT since the pole buds and somatic nuclei are still in a common cytoplasm until the end of nc10 when the pole cells cellularize. After this initial decrease, *nos* and *pgc* levels remain relatively constant until nc14, when they drop dramatically, by 80% and 54%, respectively (Fig 3B–3D). In contrast, *CycB* levels remain constant throughout this period (Fig 3B and 3E).

Because *CycB* is more abundant in the germ plasm than *nos* or *pgc* (approximately 1.8-fold and 6.5-fold, respectively) [38], *CycB* might appear to be stable if a constant rate of mRNA decay depletes the larger pool of *CycB* more slowly. Decreasing *CycB* levels by 25% at nc10-11 using a chromosomal deficiency (S2A Fig) has no effect on protection of *CycB* (S2B Fig), however. Conversely, we evaluated the effect of increasing *nos* levels using a *nos-egfp* transgene [39]. Whereas the total *nos* RNA level in nc10-11 *nos-egfp* embryos is 1.6-fold higher than in the wild type (Fig 3F), the fraction of *nos* remaining by nc14 is indistinguishable (Fig 3G). Together, these results support the conclusion that *CycB* is selectively stabilized.

To determine how long *CycB* remains stable, we quantified mRNA levels in the pole cells throughout their migration, until the pole cells coalesce in the gonad at stage 14. Although *CycB* levels do decrease significantly at stage 8, the magnitude of this decrease is much less than that of *nos* and *pgc* (34% for *CycB* versus 85% for *nos* and *pgc*) (Fig 3H–3J). After stage 8, *nos* and *pgc* levels continue to decline to 3% and 0.25% of their stage 4 levels, respectively (Fig 3H and 3I), while *CycB* levels remain steady until stage 12 (Fig 3J). The small but significant increase in *CycB* at stage 12 may result from zygotic transcription, suggesting that zygotic

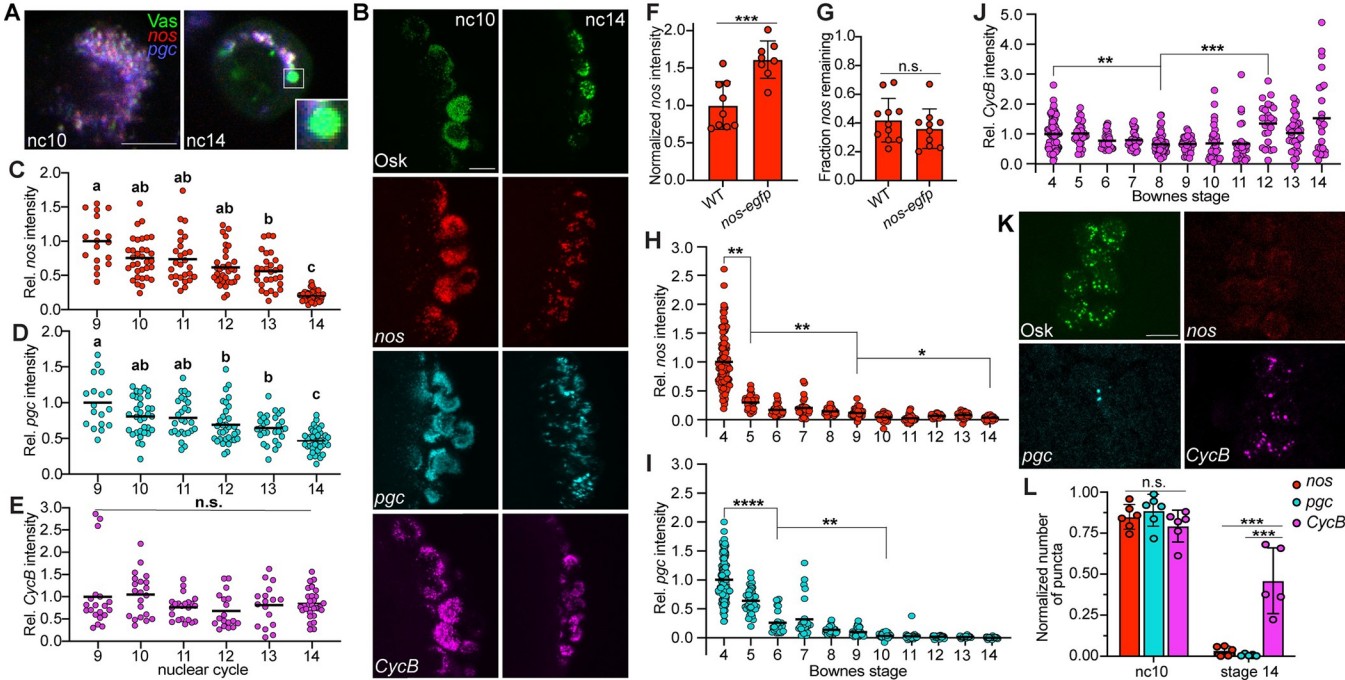

**Fig 3. *CycB* mRNA is protected while *nos* and *pgc* mRNAs are degraded.** (A) Single confocal sections of a single representative pole cell at nc10 and nc14. *nos* (red) and *pgc* (blue) were detected by smFISH. Detection of direct fluorescence of Vas-EGFP (green) was used to mark germ granules. Enlargement shows an individual granule containing Vas protein without any *nos* or *pgc* at nc14. (B) Maximum intensity confocal z-projections of representative pole cells in nc10 and nc14 embryos. *nos* (red), *pgc* (cyan), and *CycB* (magenta) mRNAs were detected by smFISH. Anti-GFP immunofluorescence (Osk-sfGFP) or direct fluorescence of Vas-EGFP was used to mark germ granules (green) and detect protein levels. (C–E) Quantification of the fluorescence intensities of *nos* (C), *pgc* (D), and *CycB* (E) per pole cell relative to the fluorescence intensity of Osk or Vas at each nuclear cycle after pole cell budding begins. Values were normalized to nc9, $n$ = 4–9 embryos per nc. Nuclear cycles that do not share a letter are significantly different as determined by Kruskal–Wallis one-way ANOVA with Dunn's post hoc test ($p < 0.05$). (F, G) Quantification of total *nos* intensity in the germ plasm in wild-type and *nos-egfp* embryos at nc10-11 (F) and nc14 (G). Values were normalized to the wild type (F) and to nc10-11 values from the same genotype (G), respectively, $n$ = 8–11 embryos per genotype. *nos* levels were compared by Student's $t$ test. (H–J) Quantification of the fluorescence intensities of *nos* (H), *pgc* (I), and *CycB* (J) per pole cell relative to the fluorescence intensity of Osk or Vas at each Bownes stage from pole cell formation to the end of pole cell migration. Values were normalized to stage 4, $n$ = 4–24 embryos per stage. mRNA levels were compared by Kruskal–Wallis one-way ANOVA and Dunn's multiple comparison test. (K) Maximum intensity confocal z-projections of representative pole cells in stage 14 embryos. Osk (green), *nos* (red), *pgc* (cyan), and *CycB* (magenta) were detected as in (B). (L) Quantification of the number of *nos*, *pgc*, and *CycB* puncta relative to the number of germ granules in nc10 and stage 14 embryos, $n$ = 5–6 embryos per stage. One-way ANOVA with Tukey–Kramer post hoc tests were performed at each time point to compare the 3 mRNAs. Individual data points and mean (C–E, H–J) or mean ± SD (F, G, L) are shown. *$p < 0.05$, **$p < 0.01$, ***$p < 0.001$, ****$p < 0.0001$, and n.s., not significant. Source data for the graphs in Fig 3C–3J and 3L are provided in S1 Data. Scale bars: 10 μm. smFISH, single-molecule fluorescence in situ hybridization.

transcripts can accumulate in germ granules. Due to the selective protection of *CycB* throughout embryogenesis, a greater fraction of germ granules at stage 14 contain *CycB* compared to *nos* and *pgc*, despite these mRNAs occupying the same fraction of germ granules when the pole cells initially form (Fig 3K and 3L).

## The mRNA decay machinery is sequentially recruited to germ granules

The loss of *nos* and *pgc* could result from their selective degradation within the germ granules, or from their selective release and subsequent degradation in the cytoplasm. To distinguish between these two possibilities, we visualized proteins involved in each major step of the 5′ to 3′ mRNA decay pathway by immunofluorescence during the period when *nos* and *pgc* degradation begins. CCR4, a component of the CCR4-NOT deadenylation complex, forms puncta that do not colocalize with the germ granules at any point during nc9 to nc14 (S3 Fig), suggesting that deadenylation is not occurring in the germ granules at these stages and may have preceded pole cell formation. At nc10, DCP1, an activating subunit of the mRNA decapping

complex, and Pacman (Pcm), the *Drosophila* homolog of the 5′ to 3′ XrnI exonuclease, form puncta in the pole cells that do not overlap with germ granules (Fig 4A and 4B). However, colocalization of germ granules with DCP1 can be detected beginning at nc12. At this time, 1 to 2 germ granules per pole cell appear to colocalize with DCP1 (Fig 4A). Interestingly, this initial colocalization occurs at the same nuclear cycle when larger germ granules first appear (Fig 1D). The overlap between DCP1 and germ granules further increases in nc13 and nc14 such that 30% of granules associate with DCP1 (Fig 4A and 4C). Pcm follows a similar pattern, but its recruitment to germ granules is delayed by 1 nuclear cycle relative to DCP1. Pcm is first detected in a few germ granules per pole cell at nc13 (Fig 4B). Colocalization increases at nc14, when approximately 30% of granules are associated with Pcm (Fig 4B and 4D). The finding that a decapping co-factor and the Pcm exoribonuclease associate with germ granules just before *nos* and *pgc* levels decrease suggests that germ granules play a role in promoting mRNA degradation in pole cells, which contrasts with their stabilizing role in early embryos.

## DCP1 is not recruited to homotypic clusters within germ granules

The presence of mRNA decay pathway proteins in germ granules raises the question of how *nos* and *pgc* can be targeted for degradation, while *CycB* RNA in the same granules is not. The organization of granule mRNAs in homotypic clusters suggests that proteins involved in the mRNA decay pathway may be selectively recruited to clusters of some RNAs, but not to others. We therefore performed stimulated emission depletion (STED) microscopy to visualize the distribution of DCP1 in relation to RNAs within germ granules before and after germ granules increase in size. Before pole cell formation, we detected *nos*, *pgc*, and *CycB* in spatially distinct homotypic clusters (S4A Fig), as has been previously described in the germ granules of early embryos [18,19]. By contrast, no separation of probes was detected when 2 differentially labeled probes for *nos* were used simultaneously. Occasionally, up to 2 homotypic clusters of *nos* and/or *pgc* were observed in the same granule (S4A Fig). In the larger granules at nc14, *nos*, *pgc*, and *CycB* remain confined to non-overlapping puncta (Figs 5A and 5B, S4B). On average, these puncta are brighter than the single transcripts detected in the bulk cytoplasm (S4C Fig), as has been found for clusters in earlier germ granules [18,19], suggesting that they contain multiple copies of the same transcript and are therefore homotypic clusters. Individual granules contain multiple clusters of *CycB*, *pgc*, and *nos*, with the number of distinct puncta per granule ranging from 2 to 12. This increase in the number of clusters in larger granules is consistent with granule growth through fusion. Surprisingly, in both large and small granules, most DCP1 puncta are spatially separated from *nos* and *CycB* during nc13 and nc14 (Figs 5C and S4D). Therefore, DCP1 localization to homotypic clusters is not required for degradation of germ granule mRNAs.

An alternate hypothesis is that the levels or activity of decapping proteins are limiting, causing the decapping complex to preferentially target mRNAs with higher binding affinity for the complex or a decapping regulatory factor. To test if DCP1 levels are limiting, we overexpressed DCP1. To avoid potential effects of excess DCP1 on germ granule assembly during oogenesis, we fused the *DCP1* coding region to the *smaug* (*smg*) 3′UTR, which inhibits *smg* translation until egg activation [40] (S5A–S5C Fig). *DCP1-smg3′UTR* RNA was expressed under GAL4/UAS control using a strong maternally active GAL4 driver, *matα-GAL4*. STED analysis of embryos overexpressing DCP1 showed that within individual germ granules, DCP1 still accumulates in distinct puncta. While many of these puncta are separated from *nos* and *CycB*, there is greater overlap between DCP1 puncta and *CycB* as compared to the wild type (Fig 5C and 5D).

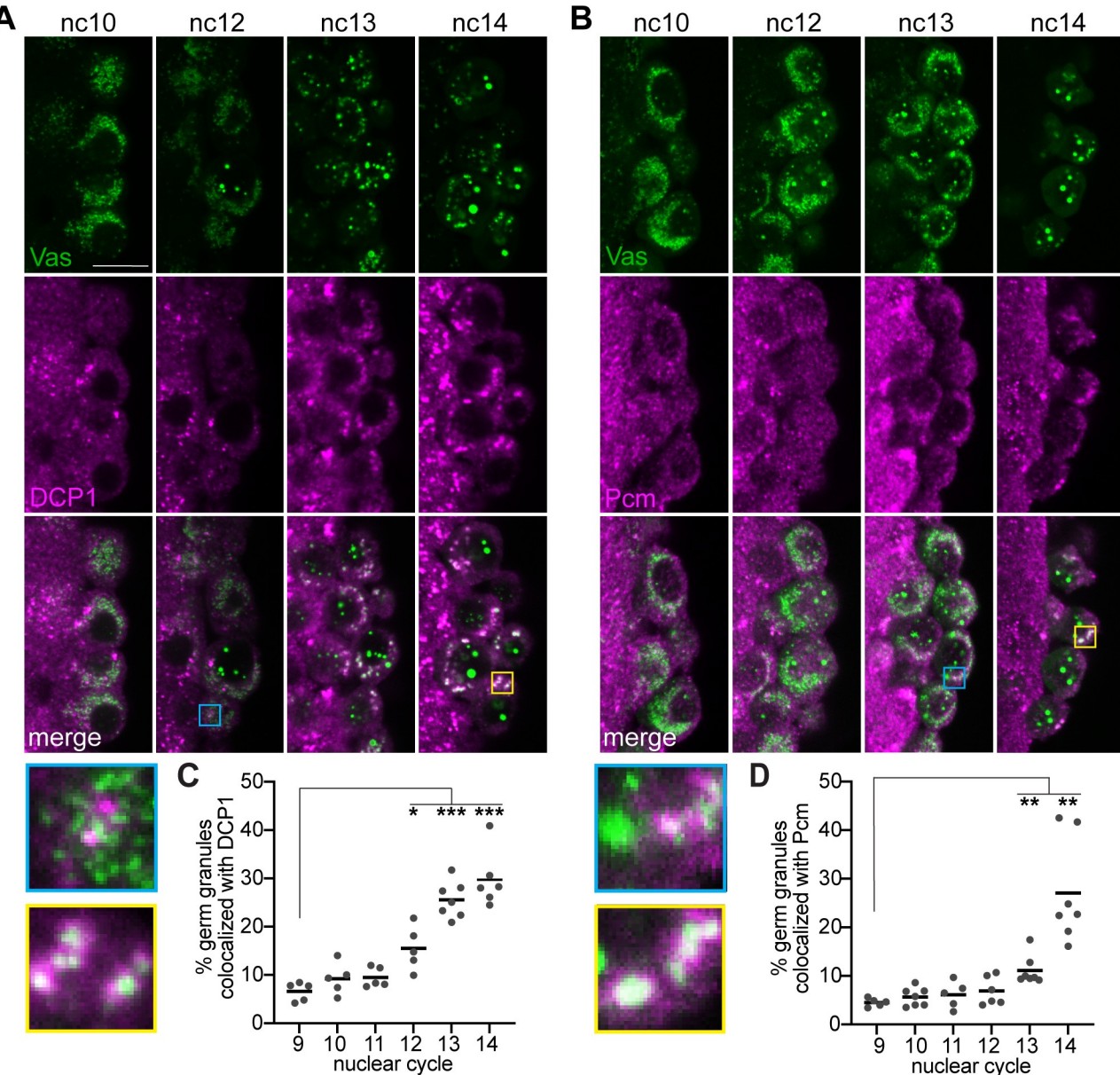

**Fig 4. mRNA decapping proteins and degradation factors localize to germ granules prior to mRNA degradation.** (A, B) Single confocal sections of the posterior region of syncytial blastoderm stage embryos expressing a *vas-efgp* transgene to mark the germ granules. Vas-EGFP (green) was detected by direct fluorescence together with anti-DCP1 immunofluorescence or anti-Pcm immunofluorescence (magenta). Enlargements of the boxed regions show examples of the earliest germ granule colocalization detected at nc12 or 13 (blue) and the strong colocalization at nc14 (yellow) for DCP1 (A) or Pcm (B). The percent of cytoplasmic Vas puncta that colocalize with DCP1 (C) and Pcm (D) puncta was quantified at each nc, *n* = 5–7 embryos per nc. Nuclear Vas puncta were masked using Imaris software. Values for individual embryos and means are shown. *$p < 0.05$, **$p < 0.01$, ***$p < 0.001$ as determined by Welch ANOVA with Dunnett's T3 post hoc test. Source data for the graphs in Fig 4C and 4D are provided in S1 Data. Scale bar: 5 μm.

To determine the effect of DCP1 overexpression on *CycB* stability, we analyzed *CycB* levels in the pole cells within the gonad by smFISH. In order to maximize the difference in DCP1 levels, we compared embryos expressing *DCP1-smg*3′UTR to embryos heterozygous for a null *DCP1* allele [41]. In *DCP1-smg*3′UTR embryos, the initial level of *CycB* in the pole cells is unchanged (S5E Fig). However, there is a small but significant reduction in *CycB* levels at nc14 (S5F Fig), and a greater decrease in the gonad (Fig 5E and 5F) when compared to *DCP1*

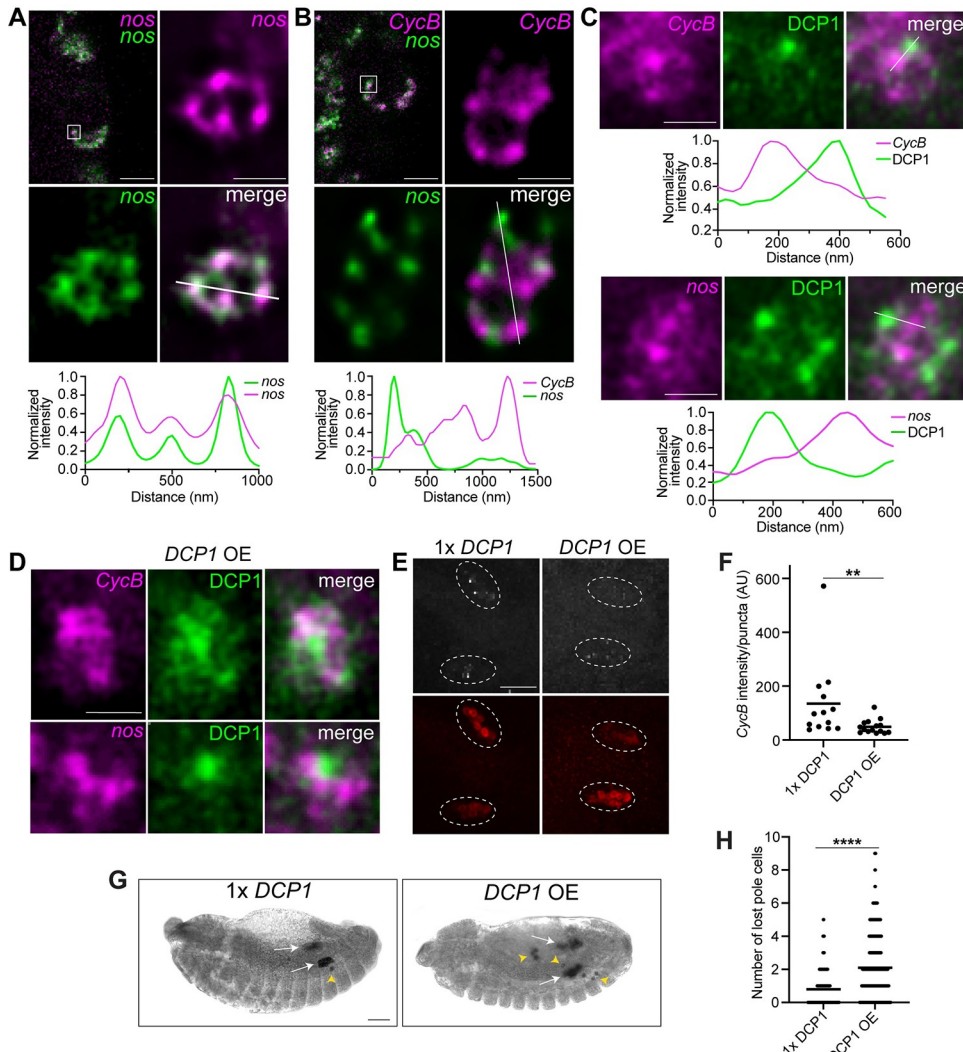

**Fig 5. DCP1 localizes to puncta within germ granules that do not overlap with *CycB* or *nos*.** (A, B) The 2D STED images of individual germ granules (indicated by the white box on the confocal section shown in the upper left panels) from a pole cell at nc13 or 14: *nos* was detected using 2 different smFISH probe sets (green + magenta) in (A); *nos* (green) and *CycB* (magenta) were detected by smFISH in (B). Fluorescence intensity was measured along the path marked with a white line and intensity profiles of each channel, normalized to the maximum value, are plotted. (C) The 2D STED images of individual germ granules showing the distribution of DCP1 (green) relative to *nos* or *CycB* (magenta) in wild-type embryos. Fluorescence intensity profiles along the path indicated by the white lines are shown. (D) The 2D STED images of individual germ granules showing the distribution of DCP1 (green) relative to *nos* or *CycB* (magenta) in embryos overexpressing *DCP1-smg3′UTR*. In all images, DCP1 was detected by immunofluorescence. STED images were deconvolved using NIS-Elements software and the brightness and contrast were adjusted individually for each image in order to best show the distributions of the mRNAs or protein at that stage. (E) Maximum intensity confocal z-projections of *CycB* detected by smFISH (white), and Vas detected by anti-Vas immunofluorescence (red) in the gonads of embryos heterozygous for a *DCP1* mutation (1× *DCP1*) and *DCP1-smg3′UTR* overexpression (*DCP1* OE) embryos. White circles outline the regions of the gonads. (F) The average fluorescence intensity of *CycB* puncta per embryo was quantified using Imaris, *n* = 13–15 embryos per genotype. (G) DIC images of representative 1× *DCP1* and *DCP1* OE embryos. Pole cells were detected by anti-Vas immunohistochemistry. The gonads (white arrows) and lost pole cells (yellow arrow heads) are indicated. (H) Quantification of the number of lost pole cells per embryo at or after Bownes stage 14 from experiment in (G), *n* = 73–223 embryos per genotype. Individual data points and mean values are shown. **$p < 0.01$, ****$p < 0.0001$ according to Mann–Whitney test. Scale bars: 5 μm for confocal images, 500 nm for STED images (A–D); 50 μm (E, G). Source data for the graphs Fig 5A–5C, 5F and 5H are provided in S1 Data. smFISH, single-molecule fluorescence in situ hybridization; STED, stimulated emission depletion.

heterozygotes. By contrast, overexpression of DCP1 does not affect the stability of *hsp83*, a pole cell enriched RNA that resides outside of germ granules (S5G Fig). Both the initial *hsp83* levels at nc10-11 and the levels at nc14 are comparable between *DCP1-smg3′UTR* and *DCP1* heterozygous embryos (S5H and S5I Fig), suggesting that DCP1 overexpression does not cause increased RNA degradation outside of the germ granules.

Notably, DCP1 overexpression causes pole cell migration defects, with an increased number of "lost" pole cells that fail to reach the gonad (Fig 5G and 5H). Since *DCP1-smg3′UTR* RNA is translated throughout the embryo (S5A Fig), this effect on pole cell migration could be due to excess DCP1 in the soma or in the pole cells. To distinguish between these possibilities, we fused *DCP1* to the *nos3′UTR*, which targets DCP1 synthesis to the germ plasm (S5A–S5C Fig). Similarly to *DCP1-smg3′UTR* embryos, DCP1 is more diffuse within the granules of *DCP1-nos3′UTR* embryos, *CycB* is aberrantly degraded after the pole cells form, and there is an increased frequency of lost pole cells (S5D, S5E and S5J–S5L Fig). This migration defect indicates that the protection of one or more germ granule mRNAs is likely important for proper pole cell function, although an indirect effect on pole cell migration due to excess DCP1 in the posterior soma cannot be completely ruled out.

## Edc3 and Patr-1 promote recruitment of the decapping complex to germ granules

The observation that mRNA decay factors are recruited to the germ granules, beginning with the decapping complex at nc12 raises the question of why and how they are recruited after pole cell formation. To better understand the mechanism of this recruitment, we knocked down the expression of known regulators of mRNA stability by RNAi expressed using *matα-GAL4* and evaluated the effect on DCP1 localization to germ granules marked with Osk-sfGFP. The efficacy of the RNAi was confirmed by RT-qPCR (S6A Fig). Consistent with the finding that CCR4 does not localize to germ granules during this period, knockdown of *twin*, which encodes CCR4 does not affect DCP1 recruitment (S6B Fig). Neither does knockdown of *pan2*, which encodes the enzymatic subunit of the Pan2-Pan3 deadenylase complex (S6B Fig). By contrast, knockdown of two decapping regulators, *edc3* and *patr-1* (also called *hpat*), leads to a decrease in colocalization of germ granules with DCP1 at nc13-14 by 29% and 19%, respectively (Fig 6A and 6B). DCP1 levels are unaffected (S6C Fig), suggesting impaired recruitment of the decapping complex. Since Edc3 and Patr-1 share several binding partners [42,43], we hypothesized that they play partially redundant roles in recruiting the decapping complex to germ granules. Consistent with this hypothesis, simultaneous knockdown of *edc3* and *patr-1* decreases the frequency of germ granule colocalization with DCP1 by 46% (Fig 6A and 6B), without affecting DCP1 levels (S6C Fig). Interestingly, *edc3+patr-1* knockdown also appears to reduce granule size at nc14 (Fig 6A), suggesting that decapping activity may be required for germ granule growth.

Since Edc3 and Patr-1 can each bind to several components of the decapping complex [42,43], we sought to determine if Edc3 and Patr-1 recruit DCP1 to the germ granules as part of the decapping complex. Therefore, we asked whether Edc3 and Patr-1 localize to germ granules during the period of recruitment. Colocalization of germ granules with Edc3 puncta was not detected (Fig 6C), suggesting Edc3 is not a component of the decapping complexes being recruited to the germ granules. Therefore, its effect on recruitment is likely indirect. By contrast, a few germ granules per cell colocalize with Patr-1 at nc12, followed by further increases at nc13 and nc14 such that 30% of granules become associated with Patr-1 (Fig 6C and 6D). This pattern of localization closely mirrors that of DCP1 (Fig 4A and 4C), supporting the

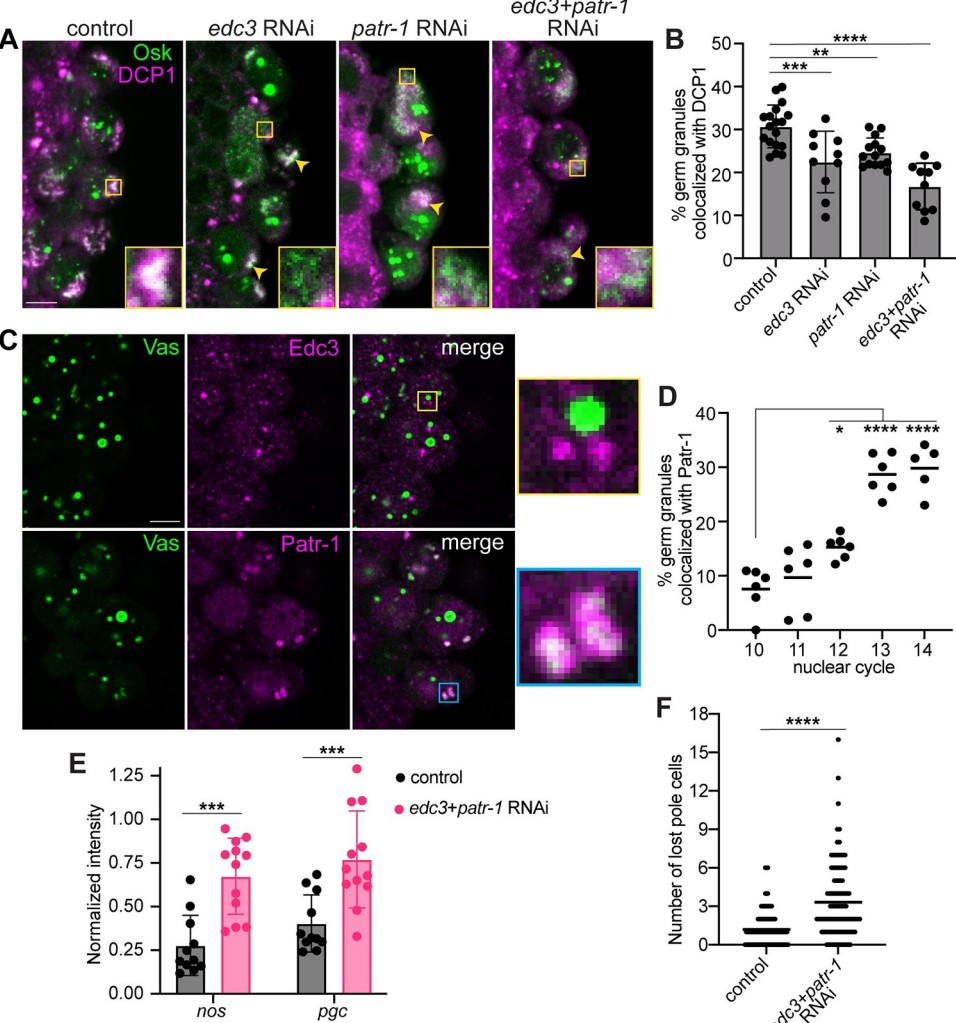

**Fig 6. Edc3 and Patr-1 promote recruitment of the decapping complex to germ granules.** (A) Single confocal sections of pole cells in nc13-14 *matα-GAL4* only, *edc3* RNAi, *patr-1* RNAi, and *edc3* and *patr-1* double RNAi embryos expressing an *osk-sfgfp* transgene. Osk-sfGFP was detected by direct fluorescence (green) together with anti-DCP1 immunofluorescence (magenta). Enlargements of the boxed regions show germ granules that recruit DCP1 in control embryos and granules that fail to recruit DCP1 in RNAi embryos. Yellow arrows indicate germ granules in the RNAi embryos that recruit Osk. (B) Quantification of the percent of cytoplasmic Osk-GFP puncta that colocalize with DCP1 in control and RNAi embryos. Genotypes were compared by Ordinary one-way ANOVA and Dunnett's multiple comparison test, *n* = 10–18 embryos per genotype. (C) Single confocal sections of the pole cells at nc14 in embryos expressing Vas-GFP (green). Vas-GFP (green) was detected by direct fluorescence and Edc3 and Patr-1 (magenta) were detected immunofluorescence. Enlargements of the boxed regions show Patr-1 (cyan box), but not Edc3 (yellow box) puncta, overlap with germ granules. (D) The percent of germ granules, marked by cytoplasmic Vas-GFP, that colocalize with Patr-1 was quantified from nc10 to nc14. Nuclear cycles were compared by Ordinary one-way ANOVA and Dunnett's multiple comparison test, *n* = 5–6 embryos per nuclear cycle. Nuclear puncta of Osk (B) or Vas (D) were masked using Imaris software. (E) The proportion of *nos* and *pgc* remaining in the pole cells at nc14 was quantified in control and double RNAi embryos. *nos* and *pgc* were detected by smFISH and their total intensities at nc14 were normalized to their average intensities during nc9-13. Genotypes were compared by Mann–Whitney test, *n* = 11–12 embryos. (F) Pole cells were detected by Vas immunohistochemistry in control and double RNAi embryos. The lost pole cells at Bownes stages 13 and later were counted, *n* = 103–106 embryos per genotype. Genotypes were compared by Mann–Whitney test. Individual data points and mean values (D, F) or mean ± SD (B, E) are shown; \**p* < 0.05, \*\**p* < 0.01, \*\*\**p* < 0.001, \*\*\*\**p* < 0.0001. Source data for the graphs Fig 6B and 6D–6F are provided in S1 Data. Scale bars: 5 μm. smFISH, single-molecule fluorescence in situ hybridization.

hypothesis that Patr-1 functions as part of the decapping complex to promote DCP1 recruitment to germ granules.

One known binding partner of Patr-1, the conserved DEAD-box helicase Me31B [42], is a component of the germ granules in the oocyte and newly fertilized embryo [44,45]. To determine if Patr-1 could promote granule localization through its interactions with Me31B, we investigated whether Me31B localizes to germ granules in the pole cells. Prior to and during pole cell budding, Me31B is present throughout the posterior of the embryo, but does not appear to be associated with germ granules (Fig 7). At nc11, Me31B accumulates at pole cell nuclei in a pattern similar to but more diffuse than that of Vas, consistent with enrichment in

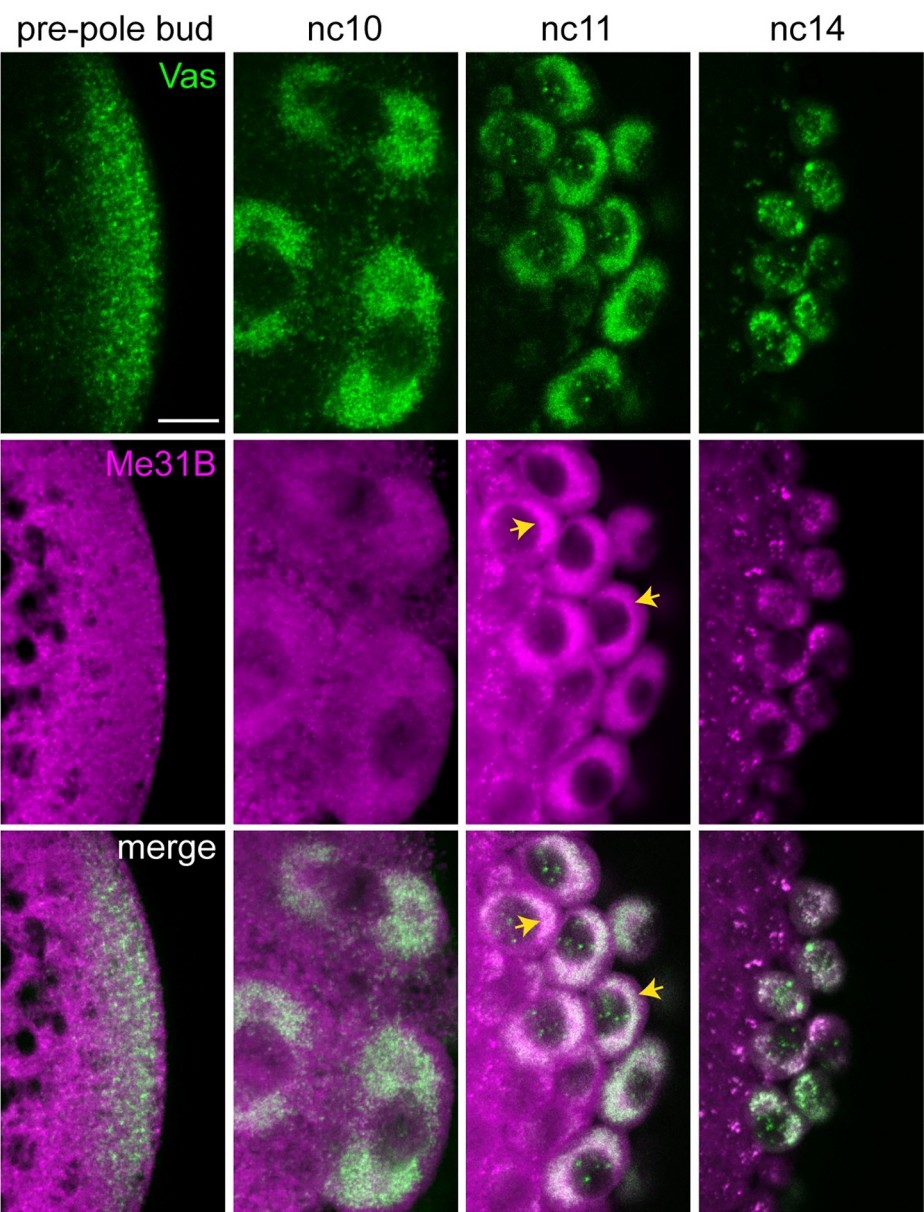

**Fig 7. Enrichment of Me31B in germ granules appears prior to Patr-1 enrichment.** Single confocal sections of the posterior region of syncytial blastoderm stage embryos expressing *Me31B-gfp* and *vas-ko* transgenes. Vas-KO and Me31B-GFP were detected by direct fluorescence. Yellow arrows indicate the Me31B-GFP signal. Scale bar: 10 μm.

germ granules. This enrichment persists into nc14 (Fig 7). Although the diffuse signal precludes quantification, Me31B appears to accumulate in germ granules before Patr-1 (Fig 6C and 6D) and DCP1 (Fig 4A and 4C) and throughout their recruitment period. Thus, Me31B could act upstream of Patr-1 to promote DCP1 recruitment. However, because oogenesis is arrested in *me31B* mutants [46], this role could not be directly tested.

## Recruitment of decapping factors to germ granules requires translation

Our results suggest that there is a temporally regulated, ordered recruitment of the mRNA decay machinery to germ granules. We therefore sought to determine the nature of the trigger that initiates this recruitment. Because the pole cells are transcriptionally quiescent during this time [47], the initiation of degradation in germ granules must be triggered by a protein translated during this period and/or posttranslational modification of a protein in the pole cells. To distinguish between these possibilities, we tested the dependence of DCP1 recruitment to the germ granules on translation, by injecting the translational inhibitor cycloheximide (CHX) into the posterior of the embryos prior to nc12 and monitoring DCP1 distribution. Although CHX can cause dissociation of P bodies, a previous study showed that they are resistant to CHX at this stage [48] and similarly, we detected DCP1 puncta in both the soma and pole cells after CHX injection (Fig 8A). Moreover, CHX injection does not affect the overall DCP1 level (Fig 8B). However, DCP1 fails to localize to germ granules in CHX injected embryos (Fig 8A and 8C). Therefore, recruitment of DCP1 to germ granules depends on translation.

One possible explanation for this observation is that translation of germ granule mRNAs makes them vulnerable to DCP1 binding and degradation, such as through gradual shortening of the poly(A) tail. This hypothesis is consistent with the protection of *CycB* RNA, as *CycB* is translationally repressed during this period [49,50]. However, in *nos* mutant embryos, where *CycB* is aberrantly translated in the pole cells during nc14 [49,50], *CycB* levels remain stable (S7A and S7B Fig). Furthermore, a *nos* RNA that cannot bind translational repressors and produces excess Nos protein [51] is slightly more stable than wild-type *nos* (S7C Fig). Therefore, translational activity is not sufficient to target germ granule RNAs for degradation. Alternatively, translation may be required to generate a critical threshold of a regulatory protein, in much the same way that synthesis of Smg has been proposed to time events in the soma at the MZT [52]. *patr-1* and *edc3* transcripts are present in pole cells [24] suggesting that accumulation of Edc3 and/or Patr-1 could trigger the recruitment of DCP1 to the germ granules. Immunofluorescence analysis showed that there is no significant difference in Edc3 or Patr-1 in the pole cells before and after nc12, however (S8A and S8B Fig). Similarly, Me31B levels are unchanged (S8C Fig). Therefore, the timing of decapping complex recruitment does not appear to be regulated by synthesis of Edc3, Patr-1, or Me31B. Rather Edc3, Patr-1, and potentially Me31B, act downstream of the trigger to promote efficient recruitment.

## Edc3 and Patr-1 are necessary for proper pole cell development

Finally, we investigated the functional importance of DCP1 recruitment to germ granules in pole cells. Because oogenesis is arrested in *DCP1* mutants [41], we could not test a requirement for DCP1 directly. Therefore, we took advantage of the defect in DCP1 recruitment in *edc3 +patr-1* double-knockdown embryos to ask whether DCP1 recruitment is required for germ granule mRNA degradation in pole cells. Quantitative smFISH analysis showed that *edc3 +patr-1* RNAi does not affect germ plasm assembly prior to pole cell formation (S6D and S6E Fig) or *nos* and *pgc* levels in the pole cells prior to nc14 (S6F and S6G Fig). At nc14, *nos* and *pgc* drop to 28% and 40% of their pre-nc14 levels, respectively, in control embryos. By contrast, 67% of *nos* and 77% of *pgc* remain intact at nc14 in double RNAi embryos (Fig 6E), suggesting

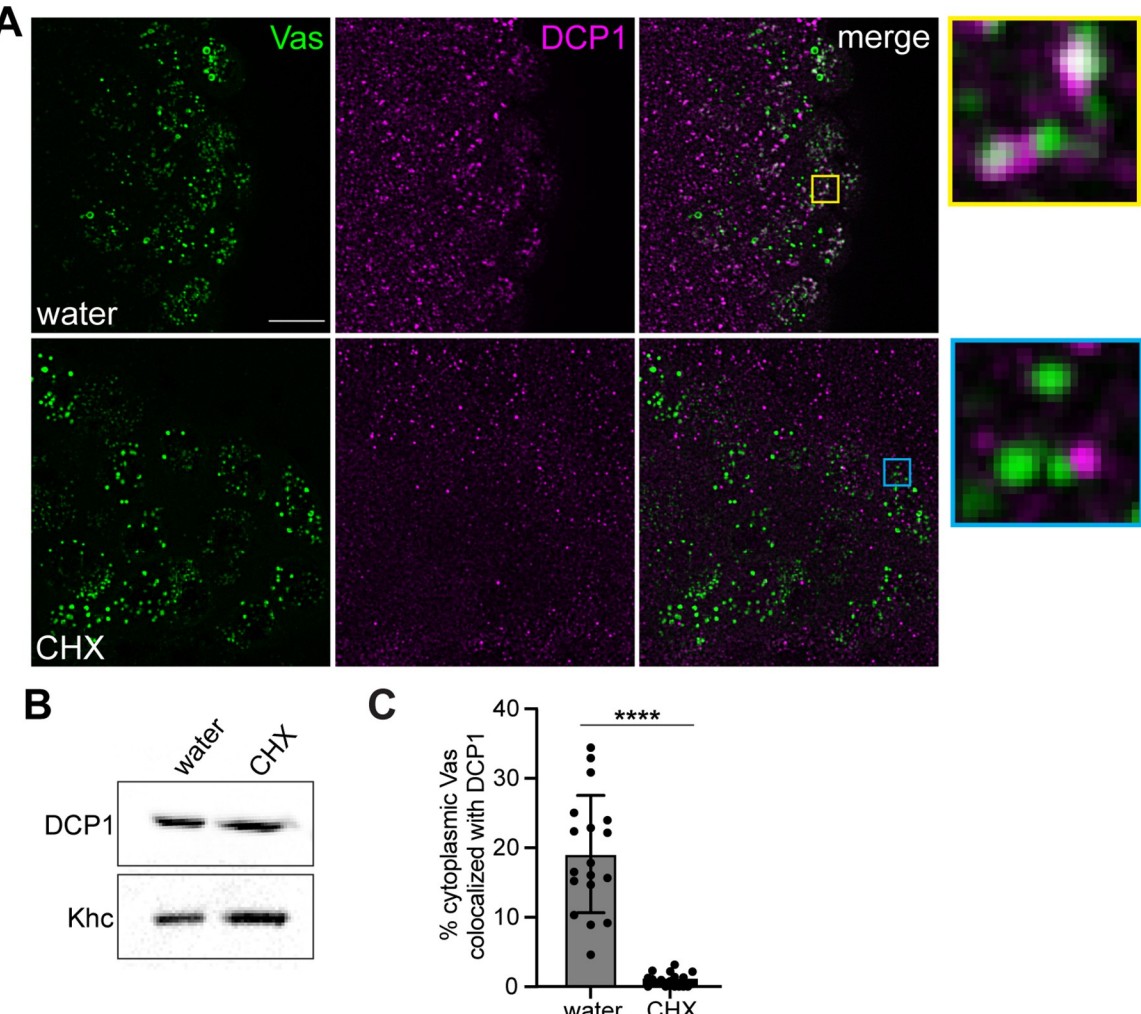

**Fig 8. Recruitment of decapping factors to the germ granules is dependent on translation.** (A) Maximum intensity confocal z-projections of the posterior of nc14 embryos expressing a *vas-egfp* transgene to mark the germ plasm after water or CHX injection. Vas-EGFP was detected by direct fluorescence together with anti-DCP1 immunofluorescence. Enlargements of the boxed regions show DCP1 localization to germ granules in water injected (control) embryos (yellow) and the lack of colocalization in CHX injected embryos (cyan). (B) Western blot analysis of DCP1 levels in CHX and water injected embryos. Khc is used as a loading control. For the unprocessed data see S1 Raw Images. (C) The percent of cytoplasmic Vas-EGFP puncta that colocalize with DCP1 was quantified in both injection conditions, *n* = 18 embryos per condition. Nuclear Vas puncta were masked using Imaris software. Individual data points and mean ± SD are shown. ****$p < 0.0001$ as determined by Welch's *t* test. Source data for the graph in Fig 7C are provided in S1 Data. CHX, cycloheximide.

that DCP1 recruitment to germ granules is necessary for *nos* and *pgc* degradation. As with DCP1 overexpression, *hsp83* was unaffected (S6H Fig), suggesting that depletion of *edc-3* and *patr-1* does not prevent RNA degradation globally in pole cells. Additionally, the stabilization of RNAs in the double-knockdown embryos allowed us to test if germ granule mRNA degradation is necessary for germline development. We found that upon *edc3+patr-1* RNAi, there is a significant increase in the frequency of lost pole cells compared to controls (Fig 6F), suggesting that recruitment of DCP1 and degradation of at least a subset of germ granule mRNAs in the pole cells is necessary for proper pole cell migration to populate the gonad.

## Discussion

Despite the ubiquity of germ granules among animal germ cells [5,6], their role in RNA metabolism is not well understood. We show that after their segregation to the pole cells, *Drosophila* germ granules undergo coordinated changes in size and function. During the period when germ granules grow by fusion, proteins involved in decapping and 5′ to 3′ degradation, but not deadenylation, are sequentially recruited to the germ granules. Therefore, deadenylation likely occurs before pole cell formation. Together, our data suggest that, in contrast to their broadly protective role in early embryos, germ granules become sites of selective mRNA decapping and degradation in pole cells and this plasticity is necessary for robust germline development.

### Role of decapping activators in DCP1 recruitment

The shift in germ granule function is promoted by two decapping activators, Patr-1 and Edc3. Patr-1 and its homologs interact with multiple proteins in the decapping complex, including the catalytic enzyme DCP2 and Me31B [42,53–55] and contribute to RNA binding [53,56]. Patr-1 localizes to germ granules 1 nuclear cycle after Me31B and concurrently with DCP1, suggesting it may serve as a direct link between the decapping complex and Me31B or mRNAs in the germ granules for recruitment of the complex. In contrast, Edc3 does not localize to germ granules, suggesting it promotes decapping complex recruitment indirectly, possibly by regulating the stability or translation of an unidentified RNA. Despite their distinct behaviors, Edc3 and Patr-1 are partially redundant as has been observed for decapping activators in other systems [55,57]. Structural and biochemical studies in yeast and *Drosophila* have shown that decapping activators such as Patr-1, Edc3, Trailer Hitch (TraI), and their homologs compete for the same binding surfaces on Me31B (Dhh1 in yeast) [42,43,55,58] and DCP2 [59], which could underlie this functional redundancy. Such redundancy would ensure effective decapping complex recruitment and mRNA degradation in pole cells.

We previously showed that founder granules, a distinct class of germ plasm RNP granules that house *osk* mRNA, also recruit mRNA decay factors [60]. This process begins much earlier, at nc5, to degrade *osk* and minimize its uptake by pole cells. During this period when germ granules and founder granules are intermingled within the germ plasm, DCP1 and Pcm associate selectively with founder granules. What makes germ granules "invisible" to the decay machinery at this time but susceptible later is an important question for future work. However, our results suggest that this susceptibility of germ granules requires production of an unidentified factor that triggers recruitment.

### Selectivity of degradation within germ granules

Our data suggest that there are two distinct class of mRNAs within the germ granules: one that is selectively targeted for decapping and degradation and one that is protected. DCP1 forms puncta in germ granules but surprisingly, DCP1 puncta do not colocalize with homotypic clusters of either class. Phase separation has been shown to repress the activity of the yeast DCP1/DCP2 complex in the absence of activators such as Edc3 [61]. Therefore, within the phase separated germ granules, these puncta could contain an inactive population of DCP1/DCP2 that cannot initiate decapping without activation. In this scenario, individual DCP1/DCP2 complexes, which would be undetectable by immunofluorescence, must exit these puncta to interact with target RNAs and become activated. Given the transient nature of enzyme–substrate interactions, DCP1/DCP2 may not accumulate at clusters. Decapping activators such as Patr-1, whose homologs recruit the decapping complex to mRNAs and catalyze cap removal [61,62] likely play a role in activating DCP1/DCP2 within the granules.

Recent work in yeast has shown that decapping activators can regulate substrate specificity. Binding of different combinations of decapping activators, including Edc3 and Pat1, to distinct *cis*-regulatory sites within the C-terminal domain of DCP2 generates distinct decapping complexes that target different sets of transcripts [57,63]. Although the specific binding sites are likely different in *Drosophila* due low conservation of the DCP2 C-terminal domain [59,64], similar interaction networks within the decapping complex occur in *Drosophila* [42,43,58]. Therefore, similar mechanisms could regulate the activity and specificity of decapping in *Drosophila*, suggesting that the selective targeting of mRNAs for degradation could be achieved by the same decapping activators that recruit DCP1 to the granules. Whether the homologs of other decapping activators that regulate substrate specificity in yeast [57,63], such as Trailer Hitch and Upf1, regulate RNA degradation in germ granules remains to be determined. By contrast, DCP1 is a general decapping activator that does not influence RNA binding [65,66]. Therefore, increasing the effective concentration of DCP1 in the granules by overexpression may be sufficient to activate decapping, but would not confer substrate specificity, leading to the loss of *CycB* in addition to *nos* and *pgc*.

## Functional significance of differential germ granule mRNA stability

Knockdown of decapping activators compromises *nos* and *pgc* RNA degradation and overexpression of DCP1 compromises *CycB* RNA protection, and both result in defective pole cell migration. This phenotype likely results from improper regulation of many germ granule mRNAs and speaks to the need for differential regulation of RNA stability. We envision that overexpression of DCP1 causes untoward decapping and degradation of mRNAs that encode proteins needed for further germ cell development and function, whereas knockdown of *edc3* and *patr1* expression results in stabilization of mRNAs and production of proteins whose functions are no longer required and may inhibit further development.

For example, *CycB* encodes a cyclin that promotes mitosis in the pole cells [26,49]. Prior to gastrulation, pole cells arrest the cell cycle in G2 through repression of CycB translation and maintain mitotic quiescence until after they reach the gonad [26,49,50]. *CycB* must be re-expressed in pole cells in the gonad for their proliferation [14]. Maintaining a pool of repressed *CycB* RNA would allow a rapid off-to-on switch, ensuring efficient entry into mitosis when the pole cells need to resume divisions in the gonad. On the other hand, *nos* and *pgc* encode proteins that delay zygotic genome activation in the pole cells [15,67] until the onset of gastrulation [24]. Pgc protein is cleared from pole cells by Bownes stage 6, which lifts global repression of transcription [15]. We find that *pgc* RNA degradation precedes protein degradation, which would limit translation and allow for effective Pgc clearance. Therefore, *pgc* degradation may promote the MZT. By contrast, Nos protein is detected in the pole cells throughout activation of zygotic transcription [47,68], suggesting repression of transcription can be lifted without Nos degradation. Therefore, the purpose of *nos* RNA degradation is unclear.

Interestingly, a recent study found that in the absence of maternal Pgc, premature miRNA transcription leads to precocious degradation of several germ granule and non-granule mRNAs in the pole cells. This zygotically driven degradation, which likely targets different RNAs than the maternal pathway described here, leads to loss of Nos, derepression of *CycB* translation, and ultimately pole cell apoptosis [69]. Therefore, the specificity and timing of mRNA degradation is likely important for pole cell development. For example, delaying degradation until nc14 could ensure global transcriptional repression is not lifted before the onset of another mechanism to silence somatic genes, such as chromatin remodeling.

## Plasticity of germ granules allows their repurposing throughout development

Our findings uncover functional plasticity of germ granules during development, with their role in localization and stabilization of maternal mRNAs in the early embryonic germ plasm supplanted by roles during pole cell formation, and then during subsequent germline development. As pole cells become less dependent on maternal mRNAs, germ granules sequentially recruit mRNA degradation proteins, which makes them more P body-like and leads to turnover of select maternal mRNAs. Recent work revealed that *C. elegans* germ granules colocalize with pre-formed P bodies and become sites of deadenylation in the germline founder cells [70]. Thus, although these species use different mechanisms to become more P body-like, the shift in function from mRNA protection to degradation may be a conserved feature of germ granules. Recruitment of new proteins provides a mechanism to regulate the function of these long-lived granules, repurposing them as needed at different developmental stages.

Interestingly, the change in germ granule function coincides with enlargement of granules by fusion. Whether this growth is a cause or effect of the functional change remains an outstanding question. An intriguing hypothesis is that germ granules are restructured to facilitate new functions. The large number of small granules facilitates distribution of the germ granule material evenly among the pole cells as they bud and divide. Once the pole cells cease division, consolidation of germ granules into fewer, large granules might be favorable by concentrating mRNAs into fewer reaction sites. Association of limiting amounts of DCP1 and Pcm with larger granules would effectively bring them into contact with a larger number of transcripts, allowing them to degrade more RNAs without having to disassociate and then reassociate with other granules. Additionally, neuronal Me31B/Imp granules increase in size and recruit new RNAs during aging, which leads to translational repression [71]. Therefore, the modulation of granule size may be a conserved mechanism to alter granule function throughout the life span of an organism.

## Materials and methods

### *Drosophila* strains and genetics

Germ granules were visualized using: endogenously *egfp*-tagged *vas* [35] or *sfgfp*-tagged *osk* (see below) genes; or *egfp-vas* (gift from R. Lehmann), *nos-Gal4::VP16*, *UAS-vasa-ko* [72], or *osk-sgfp* [73] transgenes. The following mutant and transgenic strains were used: $P_{nos}gfp$-*nos* (*egfp-nos*) [39], $dDCP1^{442P}$ [41], $nos^{BN}$ [68], *gnosSREs⁻GRH⁻* [51], *me31b-egfp* protein trap [74], *Df(2R)BSC599* (BSCD_25432), *UAS-patr-1-RNAi* (TRiP HMS01144; BDSC_34667), *UAS-edc3-RNAi* (TRiP HMS00392; BDSC_34953), *UAS-twin-RNAi* (TRiP HMS00493; BDSC_32490), and *UAS-pan2-RNAi* (TRiP GLC1808; BDSC 53249). *UAS transgenes* were expressed using *matα-GAL4-VP16* (BDSC 7062; BDSC 7063).

### Transgenic and CRISPR *Drosophila* generation

Osk was endogenously tagged at the C terminus with sfGFP or Dendra2 using the flyCRISPR scarless gene editing method (https://flycrispr.org/) [75]. The primers *osk*_gRNA_F (5′-CTTCGACGAGTCTGGAGTATTAAGT-3′) and *osk*_gRNA_R (5′-AAACACTTAATACTCCAGACTCGTC-3′) were annealed and inserted into the guide RNA plasmid pU6-BbsI-chiRNA [76] digested with BbsI. Two ApaI sites were added to the linker sequences flanking *sfGFP* within the homology-directed repair plasmid pHD-sfGFP-ScarlessDsRed to facilitate future tagging of Osk with other markers. The first site was added by PCR amplification of the entire plasmid using the primers pHD_Fwd (5′-GGGCCCTCTTGTACAGCTCATCCATG-3′)

and pHD_Rev (5′-GGGCCCTTAACCCTAGAAAGATAATCATATTGTG-3′): ApaI sites are underlined. The PCR product was digested with ApaI and self-ligated. The second ApaI site was added in the Osk1_Fwd primer described below.

The pHD-sfGFP-ScarlessDsRed+ApaI plasmid was amplified using 2 sets of primers: backbone_Fwd (5′- AAGGGCGAATTCGTTTAAACCTGCAG-3′) and backbone_Rev (5′- AAGG GCGAATTCGCGGCC-3′); linker_Fwd (5′- GGGCCCGGTGGATCTGGAGGTTCC-3′) and linker_Rev (5′- GGAACCTCCTGAACCACC-3′), and 1 kb of genomic DNA on either side of the target site in Osk were amplified from genomic DNA of flies expressing Cas9 (BDSC 78781) using the following sets of primers: Osk1_Fwd (5′- ATTTAGCGGCCGC-GAATTCGCCCTTCGTGCCCAAGATAACGGAT-3′) and Osk1_Rev (5′- AGCCGCCGGA ACCTCCAGATCCACC GGGCCC ATACTCCAGACTCGTTTCAATAACTTGC-3′); Osk2_Fwd (5′-AGGTTCTGGTGGTTCAGGAGGTTCCTAAGTTGGGTTCTTAATCAAGA TACATATATGCAA-3′) and Osk2_Rev (5′- TGCAGGTTTAAACGAATTCGCCCTTTTCC CAGTTACTTTGAACATAGCTTAGAG-3′). The 4 PCR products were joined together by Gibson assembly.

The *Dendra2* sequence was amplified from the pDendra2-C plasmid (Clontech Laboratories) with the following set of primers: Dendra2_Fwd (5′- GGGCCCGGTGGATCTGGAGG TTCCGGCGGCTCAGGGGGTAGTATGAACACCCCGGGAATT-3′) and Dendra2_Rev (5′- GGGCCCTCCACACCTGGCTGGGC-3′). The *sfGFP* sequence was excised from the pHD-Osk-sfGFP-ScarlessDsRed+ApaI homology-directed repair plasmid and replaced with Dendra2 using ApaI. Plasmids were confirmed by sequencing.

The gRNA and homology-directed repair plasmids were injected into a *nos-Cas9* line (BDSC 78781) by BestGene. Following isolation of lines with successful insertion events and removal of the Cas9 by outcrossing, flies were crossed to a *nos-PBac* line [77] to remove the DsRed cassette. Individual F1 flies that had lost DsRed expression were used to generate independent lines and the insertion was confirmed by sequencing.

For the *UASp-DCP1-smg3′UTR* and *UASp-DCP1-nos3′UTR* transgenes, the *DCP1* cDNA sequence was amplified from a *DCP1* cDNA clone (DGRC GH04763) and inserted into the pattB-UASp vector to generate pattB-UASp-DCP1. The *smg* 3′UTR and 821 bp of downstream genomic DNA was amplified from the pattB-smg5′UTR-GFP-smg3′UTR plasmid [40] using the following primers: smg_F (5′-GAATTCACCCCAATCACAACATCAACTATTTC-3′); smg_R (5′- TCTAGACTCCGTCACCAGTTGCTCTTC-3′) and inserted between the EcoRI and XbaI sites of pattB-UASp-DCP1 to generate pattB-UASp-DCP1-smg3′UTR. An EcoRI fragment containing the *nos* 3′UTR and 75 bp of 3′ genomic DNA was excised from pBS-KSnos3′UTR (Gavis lab) and inserted into the EcoRI site of pattB-UASp-DCP1 to generate pattB-UASp-DCP1-nos3′UTR. Both transgenes were integrated into the attP40 site by phiC31-mediated recombination.

### Embryo collection and fixation

Embryos were collected on yeasted apple juice agar plates at 25˚C and staged by nuclear density for nuclear cycles 9 to 14 or morphological features for Bownes stages 6 to 15. Prior to imaging, embryos were dechorionated, fixed, and devitellinized as previously described [78] and stored in methanol at −20˚C for up to 2 weeks. For western blotting or RT-qPCR, embryos were dechorionated, flash frozen in liquid nitrogen, and stored at −80˚C.

### Live imaging and photoconversion

The 2- to 3-hour-old embryos were prepared for live imaging as described in [21]. Briefly, the posterior poles of embryos were stuck to a #1.5 coverslip (VWR) using heptane glue and

embryos were covered in Halocarbon-95 oil (PolySciences). Time lapse imaging was then performed on a Nikon A1R laser scanning microscope with a resonant scanner. An approximately 8 μm z-series encompassing the pole cells was acquired using a 500 nm step size. Imaging of Osk-sfGFP was performed with a 60 × 1.4 NA oil immersion objective with a 2× optical zoom, 8× line averaging, and a capture rate of 1 z-stack per minute. Imaging of Osk-Dendra was performed with a 40 × 1.3 NA oil immersion objective with a 4× optical zoom, 4× line averaging, and a capture rate of 1 stack every 20 s. Osk-Dendra2 granules were photoconverted by 1-s stimulation with a 405 nm laser. Granules were manually tracked to identify fusion events.

### Immunofluorescence and smFISH

Embryos were rehydrated stepwise into PBST (1× PBS and 0.1% Tween-20) and incubated for 30 min in Image-iT FX (Thermo Fisher Scientific). After washing 3 × 15 min in PBHT (1× PBS, 0.1% Tween-20, 0.25 mg/mL heparin, 50 μg/mL tRNA), embryos were incubated with primary antibody in PBHT overnight at 4˚C with rocking. Primary antibodies used: 1:250 rabbit anti-CCR4 [79], 1:250 mouse anti-DCP1 [80], 1:500 rabbit anti-Pcm [81], 1:500 rabbit anti-Vas (gift from T. Schüpbach), 1:500 rabbit anti-GFP-Alexa 488 (Invitrogen), 1:500 rabbit anti-Patr-1, and 1:500 rabbit anti-Edc3 (gifts from A. Nakamura). Embryos were washed 3 × 10 min in PBHT before incubating in secondary antibody in PBHT for 2 h at room temperature with rocking. Secondary antibodies used: 1:1,000 goat anti-rabbit-Alexa 568 (Thermo Fisher Scientific), 1:1,000 goat anti-rabbit-Alexa 647 (Thermo Fisher Scientific), 1:1,000 goat anti-mouse-Alexa 647 (Thermo Fisher Scientific), and 1:250 goat anti-mouse-Abberior STAR RED. Embryos were then washed 3 × 10 min in PBST. For experiments using only immunofluorescence, embryos were incubated in 1.25 μg/mL DAPI for 2 min and rinsed 4 times in PBST. Embryos were mounted under #1.5 coverslips in Prolong Diamond Antifade Mountant (Invitrogen) for colocalization analysis and in Vectashield Antifade Mounting Medium (Vector Laboratories) for analysis of fluorescence intensity. For immunofluorescence with smFISH, embryos were re-fixed after post-secondary antibody washes in 4% PFA for 30 min before proceeding with smFISH.

smFISH was performed as previously described [78], followed by DAPI staining and mounting as described above. For embryos Bownes stage 7 and later, an additional 1 h incubation in 2% Triton-X 100 (Sigma Aldrich) was added prior to the pre-hybridization step to improve probe penetration. After staining, the late-stage embryos were cleared in RapiClear 1.47 (SUNJin Lab) overnight and mounted in 1:1 RapiClear:Vectashield.

### Confocal imaging and quantification of fluorescence intensity

Confocal imaging was performed on a Nikon A1 laser scanning microscope with a 60 × 1.4 NA oil immersion objective and GaAsp detectors, with the exception of Figs 5E and S5J, for which a 20 × 0.75 NA air objective was used. Imaging parameters were kept identical for all samples within each experiment. To quantify fluorescence intensity per pole cell (Fig 3C–3E and 3H–3J), z-series images were captured with a 300 nm step size; to quantify total fluorescence intensity of the pole cell population, a 1 μm step size was used. Images were quantified in FIJI using the "sum slices function" to capture the entire volume of a pole cell of interest and the entire pole cell population, respectively. Total fluorescence intensity (integrated density in FIJI) of the signal in the pole cell(s) and of the background signal of a region outside the embryo were then measured. Background subtracted intensities were calculated for each pole cell or embryo.

The number and intensity of puncta were analyzed using the Spots function in Imaris. Spots were detected using consistent quality thresholds within each experiment. The number

of puncta and the Intensity Sum of each spot were recorded. To quantify the normalized number of *nos*, *pgc*, and *CycB* puncta (Fig 3L), Vas, *nos*, and *pgc* or Osk and *CycB* spots were detected in the same nc10 and stage 14 images used to quantify total fluorescence intensity (Fig 3C–3E and 3H–3J). The number of RNA spots was normalized to the number of Osk or Vas spots in each embryo. Average *CycB* intensity per puncta (Figs 5F and S5K) was quantified from confocal z-series images of the whole embryo captured with 1 μm steps. Background signal was removed in FIJI using the Subtract Background function with a 2 pixel rolling ball radius and a sliding paraboloid. *CycB* signal that did not overlap with Vas (i.e., outside of the pole cells) was masked using the "Mask Channel" function in Imaris. The sum intensity of all spots was normalized to the number of spots to get an average intensity per spot for each embryo.

## Quantification of colocalization

Confocal imaging was performed using a Nikon A1 laser scanning microscope with a $60 \times 1.4$ NA oil immersion objective and GaAsp detectors. A $2\times$ optical zoom was applied to all images. An approximately 5 μm z-series encompassing the pole cells was acquired using a 300 nm step size. Prior to colocalization analysis, nuclear Vas or Osk granules were masked using the "Mask Channel" function in Imaris. The nuclear volume used for masking was defined based on DAPI signal using the surfaces function. Colocalization of Vas with DCP1 and Pcm, and of Osk and DCP1, was then analyzed as previously described [18] using a threshold of 300 nm to be considered colocalized in all analyses. Imaging conditions and thresholds were kept identical within each experiment.

## STED microscopy

For STED imaging, 1:250 goat anti-mouse STAR RED secondary antibody was used for immunofluorescence, and *nos*, *pgc*, or *CycB* probes conjugated to atto594 or atto647N were used for smFISH. Embryos were mounted in Prolong Diamond Antifade Mountant, and 2D STED imaging was performed on a Nikon A1R laser scanning microscope coupled to a STEDYCON STED module with a $100 \times 1.45$ NA oil immersion objective. Images were acquired in the STEDYCON software. A confocal image of the posterior region or pole cells was used to identify individual germ granules as the ROI for STED. STED images of a ROI within the bulk cytoplasm of early embryos were acquired from the same slides. After acquisition, STED images were deconvolved using the NIS Elements software.

To determine if germ granule puncta contain multiple transcripts, we compared the intensities of *nos*, *CycB*, and *pgc* puncta in the germ granules to the intensities of puncta in the bulk cytoplasm of early embryos, which represent single transcripts [17]. The sum intensity of each puncta was measured from deconvolved STED images using Imaris as described above. For each transcript, the intensity of each germ granule puncta was normalized to the average intensity of puncta in the bulk cytoplasm.

## Immunohistochemistry

Pole cells were detected in fixed 10- to 14-hour-old embryos using anti-Vas immunohistochemistry as previously described [60]. The following antibodies were used: 1:500 rabbit anti-Vas and 1:500 biotin goat anti-rabbit (Jackson Immuno Research Laboratories). Embryos were mounted in 80% glycerol and analyzed using a Zeiss compound microscope. Images of representative embryos were captured on a Nikon DsQi-2 microscope using a $20 \times 0.75$ NA air objective and DIC optics.

## Cycloheximide injections

Dechorionated 45- to 75-minute-old embryos were arranged end to end with the posterior poles facing the same direction and stuck to a coverslip using heptane glue. After 12 min of desiccation with Drierite (Fisher Scientific), embryos were covered in Halocarbon 200-oil (PolySciences). Embryos between 60 min and 90 min old were injected with either 1 mg/mL cycloheximide or water, at a lateral site near the posterior pole. Following injection, embryos were returned to a 25°C incubator for 70 min to develop to nc14. The embryos were removed from the coverslip by washing with heptane to dissolve the glue and rinsed thoroughly with water to remove the remaining heptane.

For western blot analysis, embryos were blotted dry, flash frozen in liquid nitrogen, and stored at −80°C. For immunofluorescence, embryos were fixed in a glass crystallization dish for 30 min. The fixative was prepared by combining equal parts heptane and a formaldehyde solution (0.5× PBS, 18.5% formaldehyde) and shaking vigorously for 1 min prior to use to saturate the heptane with formaldehyde. Several drops of the heptane and 1 drop of the aqueous phase were added to the embryos for fixation. Following fixation, embryos were hand devitellinized in PBST. Devitellinized embryos were rinsed 4× in PBST and stored in 1 mL PBST at 4°C for up to 4 days. Immunofluorescence was performed as described above.

## RT-qPCR

RNA was extracted from 0- to 2-hour-old embryos using the Qiagen RNeasy Kit. Genomic DNA removal and cDNA generation was performed with the Quantitect Reverse Transcription kit, using 750 ng total RNA per sample. Approximately 1 μl of cDNA was used for qPCR. qPCR was performed using SYBR Green PCR Master Mix (Thermo Fisher Scientific) and the StepOnePlus Real-Time PCR system (Applied Biosystems) according to manufacturer's instructions. The following primers were used at 300 nM final concentration: edc3_Fwd (5′-GCACCC TTAGAACACCACAAA-3′), edc3_Rev (5′-GGCGGTATCATGTTGCCAAA-3′), patr-1_Fwd (5′- TCGTTTTTCGGCTTTGACACG-3′), patr-1_Rev (5′-GTCATTGAGGGCATCGTATT CC-3′), twin_Fwd (5′- TGCCCACATCCGCATATACC-3′), twin_Rev (5′- TGGCTAGCCGT ATGCATCAG-3′), pan2_Fwd (5′- GGGGCCTCCGAAGACATACTA-3′), pan2_Rev (5′- GGC ACAAGCTCCACATATTCC-3′), rp49_Fwd (5′-CGGATCGATATGCTAAGCTGT-3′), and rp49_Rev (5′-GCGCTTGTTCGATCCGTA-3′). For each genotype, 2 to 3 biological replicates were performed with 3 technical replicates. Technical replicates were averaged, and the mean CT was used to calculate the expression of the target gene in each genotype relative to the *matα-GAL4* only control by the ΔΔCT method [82], using *rp49* as the reference gene.

## Western blotting

Frozen embryos were homogenized in 2 μl of sample buffer (0.125 M Tris-HCl (pH 6.8), 4% SDS, 20% glycerol, 5 M urea, 0.01% bromophenol blue, and 0.1 M DTT) per mg tissue and boiled for 4 min. Samples were then used for immunoblotting as previously described [60] using the following antibodies: 1:1,500 mouse anti-DCP1, 1:10,000 rabbit anti-Kinesin heavy chain (Cytoskeleton), 1:2,000 HRP Sheep anti-mouse, and 1:2,000 HRP donkey anti-rabbit. Blots were imaged using an iBright FL1000 Imaging System (Invitrogen).

## Supporting information

**S1 Fig. Germ granules visualized with endogenously tagged Osk or Vas proteins show same morphology and localization in pole cells throughout embryogenesis.** Maximum intensity confocal z-projections of representative pole cells at nc10, nc14, stage 9, and stage 14.

Embryos were staged by nuclear cycle or Bownes stage according to nuclear density or morphological features, respectively. Osk-sfGFP (green) was visualized by anti-GFP immunofluorescence; Vas-EGFP (green) was detected by direct fluorescence; nuclei were stained with DAPI (blue). The brightness and contrast were adjusted individually for each image to best show the features of the germ granules at that stage. Scale bar: 10 μm.
(TIF)

**S2 Fig. Reducing *CycB* levels does not affect protection of *CycB*.** (A, B) *CycB* was detected by smFISH in wild-type embryos and in embryos heterozygous for a chromosomal deficiency (*Df*) that removes *CycB*. Total *CycB* intensity in the germ plasm was quantified at nc10-11 and at nc14. (A) Total *CycB* intensity at nc10-11 normalized to wild type, $n = 8–11$ embryos per genotype. (B) Total *CycB* intensity at nc14 normalized to the average nc10-11 intensity per genotype, $n = 10$ embryos. Graphs show individual data points and mean ± SD. **$p < 0.001$ and n.s., not significant, as determined by Student's *t* test. Source data for the graphs in S2A and S2B Fig are provided in S1 Data.
(TIF)

**S3 Fig. The deadenylation complex does not colocalize with germ granules in the pole cells.** (A) Single confocal sections of the posterior region of representative syncytial blastoderm stage embryos expressing a *vas-egfp* transgene to mark the germ granules. Vas-EGFP was detected by direct fluorescence (green) together with anti-CCR4 immunofluorescence (magenta). (B) The percent of cytoplasmic Vas puncta that colocalize with CCR4 puncta was quantified at each nc, $n = 2–6$ embryos per nc. Nuclear Vas puncta were masked using Imaris software. Individual data points and means are displayed. Source data for the graph in S3B Fig are provided in S1 Data.
(TIF)

**S4 Fig. *nos*, *pgc*, and *CycB* localize to homotypic clusters in large and small germ granules.** (A) The 2D STED images of *nos*, *CycB*, and *pgc* RNA detected pairwise by smFISH in granules in pre-pole bud stage embryos. Individual granules were selected for STED imaging from confocal images as shown in the example on the left (white box). (B) The 2D STED images (indicated by the white boxes on the confocal sections shown in the left-most panels) from pole cells at nc14. *pgc* (green) was detected together with *CycB* or *nos* (magenta) by smFISH. Fluorescence intensity was measured along the paths marked with white lines and intensity profiles of each channel, normalized to the maximum value, are plotted. (C) The sum intensity of *nos* and *pgc* puncta were measured from STED images of the bulk cytoplasm of early embryos (black data points) and of clusters in single germ granules (magenta data points). Values were normalized to the average intensity of a puncta in the bulk cytoplasm. Individual data points and means are displayed. ****$p < 0.0001$ by Mann–Whitney test. Source data for the graphs in S4B and S4C Fig are provided in S1 Data. (D) The 2D STED images (indicated by the white boxes on the confocal sections shown in the left-most panels) from pole cells in nc12 and nc13 embryos comparing the distribution of *CycB* or *nos* (magenta) to the distribution of DCP1 (green). RNAs were detected by smFISH and DCP1 was detected by immunofluorescence. Scale bars: 10 μm for confocal images; 500 nm for STED images.
(TIF)

**S5 Fig. Effect of DCP1 overexpression in the germ plasm.** (A) Maximum intensity confocal z-projections of the posterior of syncytial blastoderm stage embryos that are heterozygous for a *DCP1* mutation (1× *DCP1*) or have *DCP1* overexpressed under control of the *nos* 3′UTR (*DCP1-n*) or *smg* 3′UTR (*DCP1-s*). DCP1 (green) was detected by immunofluorescence. (B) Quantification of the sum fluorescence intensity of DCP1 in the pole cells, $n = 7–14$ embryos

per genotype. (C) Western blot analysis of DCP1 levels in embryos overexpressing *DCP1* compared to *DCP1* heterozygotes. Khc was used as a loading control. For the unprocessed data, see S1 Raw Images. (D) The 2D STED images of DCP1 (green) relative to *CycB* or *nos* (magenta) in *DCP1-nos3′UTR* overexpressing embryos. (E, F) *CycB* was detected by smFISH and the total *CycB* intensity in pole cells was quantified at nc10-13, *n* = 4–10 embryos per genotype (E) and nc14, *n* = 10–11 embryos per genotype (F). Total *CycB* intensity at nc14 was normalized to the average intensity at nc10-11 (F). (G) *hsp83* (green) detected together with *CycB* (magenta) by smFISH in a wild-type nc14 embryo showing enrichment of *hsp83* in pole cells, but not in germ granules. (H) Total *hsp83* intensity in pole cells at nc10-11 (H). (I) Total *hsp83* intensity at nc14 was normalized to the average intensity at nc10-11 for the same genotype, *n* = 10–11 embryos per genotype. (J) Maximum intensity confocal z-projections of *CycB* (gray) and Vas (red) in the gonads of *DCP1* heterozygous and *DCP1-nos3′UTR* overexpressing embryos. *CycB* was detected by smFISH and Vas by immunofluorescence. (K) The average intensity of *CycB* puncta per embryo in *DCP1* heterozygous, *DCP1-nos3′UTR*, and *DCP1-smg3′UTR* (also shown in Fig 5) overexpressing embryos was measured using Imaris, *n* = 13–15 embryos. (L) Quantification of the number of lost pole cells, detected by anti-Vas immunohistochemistry in *DCP1* heterozygotes, *DCP1-nos3′UTR*, and *DCP1-smg3′UTR* (also shown in Fig 5) overexpression embryos, *n* = 73–223 embryos per genotype. Graphs display individual data points and means (E, H, K, L) or mean ± SD. (B, F, I). *$p < 0.05$, **$p < 0.01$, ***$p < 0.001$, ****$p < 0.0001$ by Kruskal–Wallis one-way ANOVA and Dunn's multiple comparison test (B, E, K, L) or Student's *t* test (F, H, I). Scale bars: 10 μm (A); 5 μm (G). Source data for the graphs in S5B, S5E, S5F, S5H, S5I, S5K and S5L Fig are provided in S1 Data.
(TIF)

**S6 Fig. Maternal RNAi reduces target RNA levels, but does not affect DCP1 levels or germ plasm assembly in early embryos.** (A) RT-qPCR analysis of *edc3*, *patr-1*, *twin*, and *pan2* RNA levels in 0–2 h embryos. The relative expression ($2^{-\Delta\Delta CT}$) of the target transcript for each RNAi was normalized to the expression in *matα-GAL4* only controls. Data are averages of biological replicates. (B) Quantification of the percent of cytoplasmic Osk-GFP puncta that colocalize with DCP1 in control, *twin* RNAi, and *pan2* RNAi embryos, *n* = 8–14 embryos per genotype. Genotypes are compared by Ordinary one-way ANOVA. (C) Western blot analysis of DCP1 levels in 0–2 h old *matα-GAL4* only, *edc3* RNAi, *patr-1* RNAi, and *edc3+patr-1* double RNAi embryos. Kinesin heavy chain (Khc) is used as a loading control. For the unprocessed data, see S1 Raw Images. (D, E) Confocal images of *nos* (magenta) and *pgc* (green) detected by smFISH in the germ plasm prior to pole cell budding in *matα-GAL4* control (D) and *edc3+patr-1* double RNAi (E) embryos. (F, G) Total *nos* (F) and *pgc* (G) intensities in the pole cells were quantified between nc9 and nc13, *n* = 9–10 embryos per genotype. (H) Total *hsp83* intensity in the pole cells was quantified at nc10 and nc14, *n* = 7–11 embryos per genotype. RNA levels in *matα-GAL4* only controls and *edc3+patr-1* double RNAi embryos were compared by Mann–Whitney test. n.s., not significant. Individual data points and mean ± SD are shown. Source data for the graphs in S6A, S6B and S6F–S6H Fig are provided in S1 Data.
(TIF)

**S7 Fig. Increased translational activity does not affect RNA stability.** (A) Maximum intensity confocal z-projections of the posterior of wild-type (WT) and *nos* mutant (*nos⁻*) embryos. *CycB* was detected by smFISH during nc9-13 and at nc14. Representative embryos at nc10 (WT), nc11 (*nos⁻*), and nc14 (both) are shown. (B) Quantification of the sum fluorescence intensity of *CycB* in the germ plasm, *n* = 10–12 embryos per time point. *CycB* levels were compared by Kruskal–Wallis ANOVA. n.s., not significant. (C) *nos* was detected by smFISH in wild-type and *gnosSREs⁻GRH⁻* embryos at nc10-11 and nc14. The *gnosSREs⁻GRH⁻* transgene

produces *nos* mRNA with mutations in binding sites for the Smaug (SREs) and Glorund (GRH) repressors (51). Total *nos* fluorescence intensity in the germ plasm was quantified and the intensity at nc14 was normalized to the average intensity at nc10-11, $n = 10–11$ embryos per genotype. *$p < 0.05$ by Student's *t* test. Individual data points and means ± SD are shown. Source data for the graphs in S7B and S7C Fig are provided in S1 Data. Scale bar: 10 μm.
(TIF)

**S8 Fig. Edc3, Patr-1, and Me31B levels do not increase in the pole cells.** (A–C) Sum intensity confocal z-projections of the posterior of nc10, nc11, and nc14 embryos. Edc3 (A), Patr-1 (B), and Me31B-GFP (C) were detected by immunofluorescence. White circles indicate the region of the pole cells. Total fluorescence intensity in the pole cell region of embryos during nc9-11 and nc13-14 (A, B) or nc10 and nc11 (C) was quantified, $n = 3–6$ embryos each. n.s., not significant by Student's *t* test. Source data for the graphs in S8A–S8C Fig are provided in S1 Data. Scale bar: 10 μm.
(TIF)

**S1 Raw Images. Unprocessed western blot image files accompanying Figs 8B, S5C, S6C and S7A.** TIF images of raw blots uncropped and labeled with antibody used and sample identifier. Lanes indicated by X's are not relevant to Figs 8B and S5C.
(PDF)

**S1 Data. Original data accompanying Figs 3–6, Fig 8 and S2–S8.** Individual values used to generate graphs for each figure are listed under the corresponding tab.
(XLSX)

**S1 Video. Germ granules grow by fusion.** Time lapse movie of a representative pole cell expressing endogenously tagged Osk-sfGFP during nc14. A 1 μm region of the pole cell is shown. Yellow arrows indicate granules before and after fusion. Images were captured at 1 z-series per minute. Time stamp indicates minutes:seconds.
(MP4)

**S2 Video. Germ granules exchange their contents during fusion.** Time lapse movie of a representative pole cell expressing endogenously tagged Osk-Dendra2 during nc14. After photoconverting a portion of the germ granules, photoconverted (magenta) and un-photoconverted (green) granules were observed for fusion. A 1 μm region of the pole cell is shown. Yellow arrows indicate the granules of interest throughout the video. Images were captured at 1 z-series per 20 s. Time stamp indicates minutes:seconds.
(MP4)

**S3 Video. Additional example of germ granules exchanging their contents during fusion.** Time lapse movie of a pole cell expressing endogenously tagged Osk-Dendra2 during nc14. After photoconverting a portion of the germ granules, photoconverted (magenta) and un-photoconverted (green) granules were observed for fusion. A 1 μm region of the pole cell is shown. Yellow arrows indicate granules before and after the exchange of their contents. Images were captured at 1 z-series per 20 s. Time stamp indicates minutes:seconds.
(MP4)

## Author Contributions

**Conceptualization:** Anna C. Hakes, Elizabeth R. Gavis.

**Formal analysis:** Anna C. Hakes.

**Funding acquisition:** Elizabeth R. Gavis.

**Investigation:** Anna C. Hakes.

**Project administration:** Elizabeth R. Gavis.

**Supervision:** Elizabeth R. Gavis.

**Writing – original draft:** Anna C. Hakes, Elizabeth R. Gavis.

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
