## [Editor Report · Decision Letter 0]

15 Nov 2022

Dear Dr Gavis, 

Thank you for submitting your manuscript entitled "Plasticity of Drosophila germ granules during germ cell development" for consideration as a Research Article by PLOS Biology.

Your manuscript has now been evaluated by the PLOS Biology editorial staff as well as by an academic editor with relevant expertise and I am writing to let you know that we would like to send your submission out for external peer review.

Once your full submission is complete, your paper will undergo a series of checks in preparation for peer review. After your manuscript has passed the checks it will be sent out for review. To provide the metadata for your submission, please Login to Editorial Manager (https://www.editorialmanager.com/pbiology) within two working days, i.e. by Nov 17 2022 11:59PM.

Kind regards,

Ines

--

Ines Alvarez-Garcia, PhD

Senior Editor

PLOS Biology

---

## [Decision Letter · Decision Letter 1]

22 Dec 2022

Dear Dr Gavis,

Thank you for your patience while your manuscript entitled "Plasticity of Drosophila germ granules during germ cell development" was peer-reviewed at PLOS Biology. It has now been evaluated by the PLOS Biology editors, an Academic Editor with relevant expertise, and by three independent reviewers. 

The reviews are attached below. As you will see, the reviewers find the conclusions interesting and significant for the field, but they also raise several issues that would need to be addressed. They all think that the imaging approach is powerful, but also limited to demonstrate molecular associations and whether or not they are direct or indirect. They offer alternative explanations to some of the findings and suggest additional analyses to discard them. After discussing the reviews with the Academic Editor, we would like to invite you to revise the work to address the reviewers' comments by either performing simple experiments to confirm the conclusions or by acknowledging clearly the limitation of the method and data obtained in the text.

Given the extent of revision needed, we cannot make a decision about publication until we have seen the revised manuscript and your response to the reviewers' comments. Your revised manuscript is likely to be sent for further evaluation by all or a subset of the reviewers.

**IMPORTANT - SUBMITTING YOUR REVISION**

3. Resubmission Checklist

a) *PLOS Data Policy*

b) *Published Peer Review*

d) *Blurb*

Please also provide a blurb which (if accepted) will be included in our weekly and monthly Electronic Table of Contents, sent out to readers of PLOS Biology, and may be used to promote your article in social media. The blurb should be about 30-40 words long and is subject to editorial changes. It should, without exaggeration, entice people to read your manuscript. It should not be redundant with the title and should not contain acronyms or abbreviations. For examples, view our author guidelines: https://journals.plos.org/plosbiology/s/revising-your-manuscript#loc-blurb

Sincerely,

Ines

--

Ines Alvarez-Garcia, PhD

Senior Editor

PLOS Biology

Reviewers' comments

Rev. 1: Akira Nakamura – note that this reviewer has signed his review

In this manuscript, Hakes and Gavis investigate the morphological transition of germ granules during Drosophila embryogenesis. They found that germ granules fuse and enlarge during pole cell formation. Concurrently, several factors involved in mRNA decay, such as DCP1, PCM, and PATR-1, are recruited to the germ granules, and nos and pgc mRNAs appear to be selectively degraded. Modulation of the activity of decay factors inhibits proper pole cell migration to the gonad. They propose that the germ granules change their shape and are repurposed for distinct functions during pole cell formation.

This is probably the first detailed report at a high resolution that germ granules fuse and change morphology during pole cell formation and subsequent migration to the embryonic gonads. The quality of mages is very high and provides a nice piece of information, which will be useful for future research. However, the results are descriptive. The authors do not provide direct proof of how the processes are operated at the molecular level. Thus, the arguments drawn by the authors are, unfortunately, not convincing. I also think that the results can be explained in other ways (see below).

Major concerns and questions:

1. Selective degradation of germ granule mRNAs: The authors observe that nos and pgc mRNAs start to be degraded during pole cell formation, whereas the CycB transcript is stable. According to RNA-seq data in the Flybase, however, the amount of CycB mRNA in 0-2 h embryos is about five-fold compared with those of nos and pgc. My personal experience of in situ hybridization with the AP-conjugated anti-DIG antibody followed by X-phosphate/NBT staining supports that CycB is the most abundant germ plasm mRNA. If the speed of degradation (e.g. molecules/time) is constant among germ granule mRNAs, the rate of decrease in the CycB mRNA level would be much slower. For example, when nos or pgc mRNA levels decreased to 5%, still >80% CycB mRNA would remain. If this is the case, there would be no selective mechanism for the degradation of germ granule mRNAs. I think that the authors need to discuss and exclude this possibility.

2. Physiological roles of selective degradation of germ granules mRNAs: It is plausible that, when a sufficient amount of nos mRNA is localized in the germ plasm, ~20% nos mRNA in stage-5 (nc14) pole cells would be enough for its function during subsequent embryogenesis (the reduction of nos and pgc mRNAs in pole cells is a physiologically normal event during their removal in the somatic region). If the degradation mechanism is turned not to be selective, I think that the authors will have to dramatically rebuild the overall logical flow in this manuscript.

3. Page 8 (absence of deadenylation in germ granule mRNAs): How about PARN? In vertebrates, PARN, which is another type of deadenylase, forms a complex with CPEB (the Drosophila Orb homolog) and GLD2, which is a cytoplasmic poly(A) polymerase. The competition between PARN and GLD2 in the CPEB complex regulates the poly(A) tail length of Cyclin-B1 mRNA in the Xenopus oocyte (Kim and Richter, Mol. Cell 2006). Although the distribution of Orb protein in early-stage embryos has not been documented at a high resolution, orb mRNA is known to be enriched in the germ plasm (Lécuyer et al., Cell 2007). The authors may be worthy of examining the localization of Orb complex during pole cell formation. Also, the poly(A) tail is often shortened in a translation-dependent manner, which triggers decapping and mRNA decay (Funakoshi et al. Genes Dev 2007).

4. Pages 10-11 (DCP1 overexpression and edc3/patr1 RNAi): The authors show that the overexpressed DCP1 accumulates in the germ granules and causes CycB mRNA degradation. Fig S4 shows that DCP1 is also present in the cytoplasm when it is overexpressed. I wonder if the overexpression of DCP1 also promotes the degradation of another class of pole cell mRNAs such as wun2 and tre1 mRNAs, which appear not to reside in the germ granules but play important roles in steering pole cell migration (Kunwar et al., PLOS Biol. 2003; Hanyu-Nakamura et al. Development 2004; Renault et al., Science 2004). A similar question is also to the edc3/patr1 RNAi, which reduces DCP1 association to the germ granules (Fig 6). Because the edc3/patr1 RNAi does not affect the total DCP1 level, the cytoplasmic DCP1 level is likely to be increased. If these non-granular mRNAs are degraded by DCP1 overexpression and edc3/patr1 RNAi, another hypothesis would be that DCP1 has to be sequestered in the separate compartments within the germ granules for the protection of maternal mRNAs in pole cells.

5. Pages 11 and 14 (defects in pole cell migration): The authors show that DCP1 overexpression (enhancement of decapping) and edc3/patr1 RNAi (diminishment of decapping) both cause defects in pole cell migration. How do the authors interpret the results from the opposite manipulations?

6. Another question for these experiments is whether the germ plasm behaviors (fusion and enlargement) are affected by DCP1 overexpression and edc3/patr1 RNAi.

Minor points

1. Pages 9-10 (The DCP1 association with the germ granules): DCP1 is a co-activator of decapping catalyzed by the DCP2 subunit. The authors may need to examine the distribution of DCP2.

2. Page 11 and Fig S4 (DCP1-nos3'UTR embryos): The DCP1 signals are still detectable in somatic cells. Thus, I think that the authors still cannot exclude the possibility that DCP1 in the soma indirectly affects pole cell migration.

3. Page 12 and Fig 7 (nuclear Me31B signals): the signals are not clear in the figure. Please point the nuclear signals with arrowheads.

4. Page 14 (edc3/patr1 knockdown) Edc3 and Patr1 are known to interact with Me31B, which is a component of the osk RNP during oogenesis (ref. 44). Is the germ plasm assembly normal in edc3/patr1 RNAi embryos?

Rev. 2:

The manuscript by Hakes and Gavis provides a detailed look at the dynamics of germ granules, three of their constituent mRNAs, and proteins involved in RNA degradation, during embryonic development from the time of germ cell specification through the coalescence of germ cells into gonads at stage 14. Consistent with previous results, they show that RNAs that are associated with germ granules are protected from degradation. However, they demonstrate that starting at around nuclear cycle 12, germ granules fuse, and that this is coupled with altered function, such that nos and pgc RNAs become targets for degradation while cycB RNA remains protected. Degradation of nos and pgc RNAs involves recruitment of the decapping factor DCP1, and this in turn is shown to be promoted by the decapping activators Edc3 and Patr-1, and apparently does not involve the deadenylase CCR4. The authors also demonstrate that overexpression of DCP1 results in destabilization of cycB RNA, indicating a loss of RNA substrate selectivity when DCP1 is in excess. Interfering with this process of selective degradation of germ granule RNAs, by overexpression of DCP1 or double knockdown of Edc3 and Patr-1, results in an increased number of pole cells that fail to migrate appropriately during embryonic development to find the gonadal mesoderm. The increases observed would not be expected to be enough to produce sterility in the adults that develop from these embryos, and in any case fertility is not directly tested. Finally, by injecting cycloheximide into germ plasm the authors demonstrate that translation is necessary for recruitment of DCP1 into germ granules, but the identity of the RNA or RNAs that need to be translated remains undetermined.

The experiments included in the paper are all very carefully performed and fully support the conclusions made by the authors. The manuscript is also very clearly presented and was a pleasure to read. However, the paper is limited by the fact that it relies solely on imaging to demonstrate molecular associations. Co-localization experiments cannot distinguish direct from indirect molecular associations and cannot provide detailed molecular information about specific protein-protein and protein-RNA interactions that are relevant to establishing which RNAs are targeted for degradation and which are protected.

In my view these comments do not preclude publication of this work in PLoS Biology, because molecular and biochemical analyses from Drosophila germ granules are not technically practical. However the work could be usefully extended by building on what has been found in yeast, where more is known about mechanisms of establishing selective degradation. Ref. 54 discusses how several different decapping activators bind to different sites in the C-terminal region of DCP2 and form different decapping complexes, which target different subsets of mRNAs. While orthologs of some of these activators were included in this study, others (Upf1, Lsm1, tral (the Scd6 ortholog) were not. Nor was DCP2 analyzed in detail. Is it at least known whether the other decapping activators are present in germ plasm and therefore could contribute to specificity? Are the binding sites for decapping activators conserved in fly DCP2? While it would be a lot to address these and other related questions experimentally, at a minimum I would like to see some additional discussion of experimental results from yeast and how they may or may not pertain to the observations made in this work.

Minor comments:

Referring to the manuscript at the end of page 8 and beginning of page 9, the distributions of DCP1 and Pcm are described as similar to that of CCR4. I think the distributions are only similar in the sense that they form puncta that do not overlap with germ granules. Lacking co-localization experiments with CCR4 I think the sentence requires clarification.

Line 14 of the abstract: should read 'differentially' not 'differential'.

On page 4, the Su et al. 1998 reference needs to be numbered (I think it is ref. 61).

On page 16, line 5, it should read 'repress' not 'represses'.

Rev. 3: Ekaterina Voronina – note that this reviewer has signed her review

In this manuscript, Hakes and Gavis investigate morphological and functional transitions of germ granules during Drosophila germ cell development. Germ granules regulate stability and translation of maternal mRNAs important for germ cell development. The authors show that germ granule enlargement at stages nc 13-14 is likely mediated by fusion of pre-existing small germ granules. Importantly, this morphological change coincides with a functional transition, where several germ granule-associated mRNAs are destabilized, while others are maintained throughout embryogenesis. This change of germ granule function is associated with recruitment of mRNA decay factors, which requires translation. Both selective protection and destabilization of germ granule mRNAs are necessary for proper germ cell development and their migration to the gonads. It is not yet clear what distinguishes the destabilized transcripts from those still maintained. This is a thorough and well-written manuscript that uncovers functional plasticity of RNA granules relevant to many biological contexts. However, there are some issues with data interpretation that need to be addressed before publication:

1) Fig. 6/Page 11 section "Edc3 and Patr-1 promote recruitment of the decapping complex to germ granules"

Fig. 6A: the assembly of large cytoplasmic Osk granules appears to be disrupted by both edc3 and patr-1 RNAi; in the close-ups, cytoplasmic germ granules of RNAi strains are smaller than the control granules at the same stage. Instead of a failure of DCP1 localization to appropriately enlarged germ granules, it seems as though disruption of decapping prevents the developmental switch in germ granule morphology and function. This possibility should at least be acknowledged.

2) Fig. 6/Page 14 section "Recruitment of DCP1 to germ granules is necessary for proper pole cell development"

The title of this section of the Results is problematic: The data does not distinguish whether the defect in edc3/patr-1 double RNAi is in fact DCP1's localization to germ granules versus DCP1's ability to be activated for decapping or to be targeted to specific mRNAs. If the embryo doesn't have patr-1/edc3, decapping will be compromised whether or not DCP1 is in the germ granules. The section title should be changed to avoid overinterpreting the results.

---

## [Decision Letter · Decision Letter 2]

3 Mar 2023

Dear Dr Gavis,

Thank you for your patience while we considered your revised manuscript entitled "Plasticity of Drosophila germ granules during germ cell development" for publication as a Research Article at PLOS Biology. This revised version of your manuscript has been evaluated by the PLOS Biology editors, the Academic Editor and one of the original reviewers.

Based on the reviews, we are likely to accept this manuscript for publication, provided you satisfactorily address the data and other policy-related requests stated below.

In addition, we would like you to consider a suggestion to improve the title, but feel free to change it if it is not accurate:

"Drosophila germ granule composition and function are differentially regulated during germ cell development"

We expect to receive your revised manuscript within two weeks. 

*Published Peer Review History*

*Press*

Sincerely,

Ines

--

Ines Alvarez-Garcia, PhD

Senior Editor

PLOS Biology

DATA POLICY:

Many thanks for for submitting the data underlying all the graphs shown in the figures. Please also ensure that all the figure legends in your manuscript include information on where the underlying data can be found.

BLURB

Please also provide a blurb which (if accepted) will be included in our weekly and monthly Electronic Table of Contents, sent out to readers of PLOS Biology, and may be used to promote your article in social media. The blurb should be about 30-40 words long and is subject to editorial changes. It should, without exaggeration, entice people to read your manuscript. It should not be redundant with the title and should not contain acronyms or abbreviations. For examples, view our author guidelines: https://journals.plos.org/plosbiology/s/revising-your-manuscript#loc-blurb

Reviewers' comments

Rev. 1: Akira Nakamura

I read the revised manuscript. I am very glad that the authors have positively responded to my comments through further experimentation and additional discussion in the text. My initial concerns have been satisfactorily addressed, and the authors' arguments are convincingly supported by their data. Thus, I strongly recommend its publication in PLOS Biology.

---

## [Editor Report · Decision Letter 3]

7 Mar 2023

Dear Dr Gavis,

Thank you for the submission of your revised Research Article entitled "Plasticity of Drosophila germ granules during germ cell development" for publication in PLOS Biology. On behalf of my colleagues and the Academic Editor, Yukiko Yamashita, I am delighted to say that we can in principle accept your manuscript for publication, provided you address any remaining formatting and reporting issues. These will be detailed in an email you should receive within 2-3 business days from our colleagues in the journal operations team; no action is required from you until then. Please note that we will not be able to formally accept your manuscript and schedule it for publication until you have completed any requested changes.

PRESS

Sincerely, 

Ines

--

Ines Alvarez-Garcia, PhD

Senior Editor

PLOS Biology
